# HPF1 and nucleosomes mediate a dramatic switch in activity of PARP1 from polymerase to hydrolase

Johannes Rudolph[1], Genevieve Roberts[1], Uma M Muthurajan[1], Karolin Luger[2]*

[1]Department of Biochemistry, University of Colorado Boulder, Boulder, United States; [2]Howard Hughes Medical Institute, University of Colorado Boulder, Boulder, United States

**Abstract** Poly(ADP-ribose) polymerase 1 (PARP1) is an important player in the response to DNA damage. Recently, Histone PARylation Factor (HPF1) was shown to be a critical modulator of the activity of PARP1 by facilitating PARylation of histones and redirecting the target amino acid specificity from acidic to serine residues. Here, we investigate the mechanism and specific consequences of HPF1-mediated PARylation using nucleosomes as both activators and substrates for PARP1. HPF1 provides that catalytic base Glu284 to substantially redirect PARylation by PARP1 such that the histones in nucleosomes become the primary recipients of PAR chains. Surprisingly, HPF1 partitions most of the reaction product to free ADP-ribose (ADPR), resulting in much shorter PAR chains compared to reactions in the absence of HPF1. This HPF1-mediated switch from polymerase to hydrolase has important implications for the PARP1-mediated response to DNA damage and raises interesting new questions about the role of intracellular ADPR and depletion of NAD$^+$.

## Introduction

DNA damage is a frequent occurrence in cells and must be continuously monitored and repaired in order to maintain genome stability (*Peterson and Côté, 2004*). The abundant nuclear proteins poly (ADP-ribose) polymerase 1 and 2 (PARP1 and PARP2) serve as early detectors of DNA damage (*Gibson and Kraus, 2012*; *Ray Chaudhuri and Nussenzweig, 2017*). PARP1, upon binding to damaged DNA, binds NAD$^+$ and transfers the ADP-ribose (ADPR) portion onto itself and other proteins. PARP1 then elongates this ADPR to form a chain in a process termed poly(ADP-ribosyl)ation (PARylation), which can take place on PARP1 (autoPARylation) or other proteins (transPARylation) (*Gibson and Kraus, 2012*; *Morales et al., 2014*; *Bai, 2015*). These PAR chains are attached to thousands of nuclear proteins with histones being the most abundantly PARylated proteins apart from PARP1 (*Bonfiglio et al., 2017a*; *Larsen et al., 2018*). More recently it has been shown that histone-linked PARylation depends on Histone PARylation Factor 1 (HPF1) (*Gibbs-Seymour et al., 2016*). Interestingly, HPF1 not only promotes transPARylation of histones but also alters the specificity of the modified residues from aspartate or glutamate to serine (*Palazzo et al., 2018*). The PAR chains attached to PARP1 and other proteins lead to the recruitment of DNA repair factors that contain PAR-binding motifs, such as macrodomains, WWE domains, or PAR-binding zinc fingers (*Teloni and Altmeyer, 2016*). It is thought that different types of PAR chains (i.e., long vs. short, branched vs. unbranched) may lead to recruitment of different repair factors in a concept termed the PAR code (*Aberle et al., 2020*; *Karlberg et al., 2013*). Deletion of PARP1 leads to increased carcinogenesis, and deletion of both PARP1 and its most closely related homolog, PARP2, is embryonically lethal (*Amé et al., 1999*; *Ménissier de Murcia et al., 2003*). PARP1 (and PARP2) are important targets for cancer therapy with now four clinically approved compounds (olaparib, rucaparib, niraparib, and

*For correspondence:
karolin.luger@colorado.edu

Competing interests: The authors declare that no competing interests exist.

talazoparib) in use for treatment of breast and ovarian cancer (*Yi et al., 2019*). Additionally, these and other inhibitors of PARP are being investigated as cancer treatments in combination with agents that damage DNA (e.g., temozolomide, cis-platinum, checkpoint inhibitors) (*Yi et al., 2019*). All clinically relevant inhibitors of PARP bind in the $NAD^+$ binding pocket of the catalytic domain of PARP1 and PARP2 (*Thorsell et al., 2017*).

From a mechanistic point of view, PARP1 is a very complicated enzyme that catalyzes four different reactions (*Alemasova and Lavrik, 2019*). In the first reaction, $NAD^+$ serves as a substrate for ADP-ribosylation of an amino acid side chain wherein the nucleophilic side chain attacks C1′′ to release nicotinamide (*Figure 1A*). A number of aspartates and glutamates in the BRCT domain of PARP1 itself are the primary sites of modification in autoPARylation reactions (*Tao et al., 2009*; *Zhang et al., 2013*; *Vyas et al., 2014*; *Gagné et al., 2015*). The lack of specificity for a particular site (or protein) has been frustrating and confusing the field for decades (*Tao et al., 2009*; *Suskiewicz et al., 2020a*). In the presence of HPF1, the specificity of PARP1 switches to serine side chains, again located primarily in the automodification regions between the end of the BRCT domain and the WGR domain for autoPARylation, and in histone tails for transPARylation (*Larsen et al., 2018*; *Bonfiglio et al., 2017b*). In the second reaction catalyzed by PARP1, $NAD^+$ is used to extend

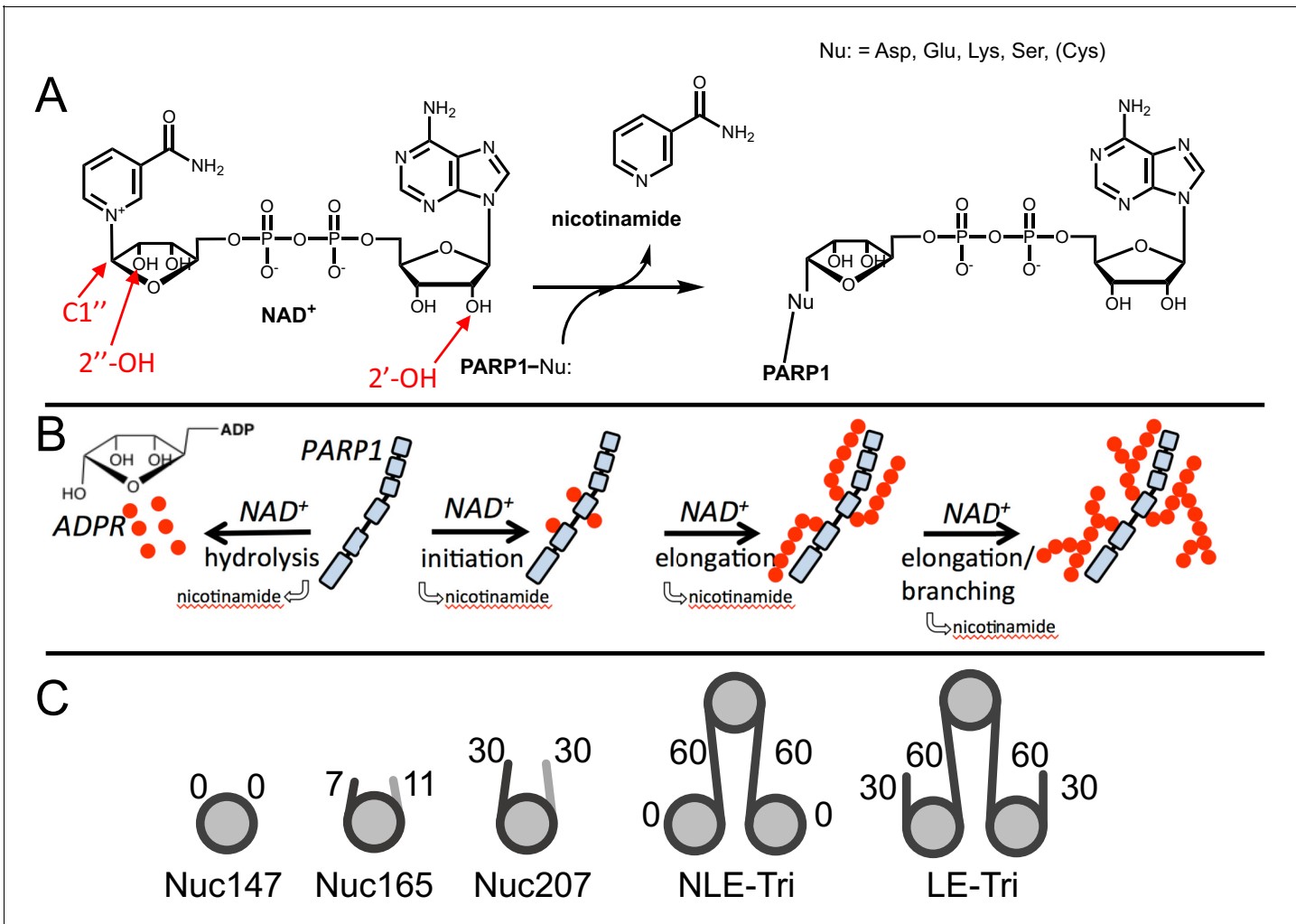

**Figure 1.** Chemical mechanism of poly(ADP-ribose) polymerase 1 (PARP1) and structures of nucleosomes used here. (**A**) Chemical mechanism of PARP1 highlighting in red important groups of $NAD^+$ as related to initiation (C-1′′), elongation (2′-OH), and branching (2′′-OH). (**B**) Cartoon representation of the four reactions catalyzed by PARP1 (in light blue): initiation, elongation, branching, and hydrolysis. ADP-ribose (ADPR) monomers (red circles) form PAR chains or free ADPR. (**C**) Cartoon representation of the different nucleosome constructs utilized in this study. The number in the name indicates the length of DNA (e.g., Nuc147 has 147 bp of DNA), and the small numbers next to the linkers or ends indicate the length of these regions.

the initially attached ADPR, wherein the 2′-OH (the acceptor hydroxyl) of this ADPR is deprotonated by Glu988 (*Ruf et al., 1998*; *Figure 1B*). Repeating this process, PARP1 builds long PAR chains of up to 200 units on itself with a large range reported for average chain lengths (*Kawaichi et al., 1981*; *Sugimura and Miwa, 1982*; *Alvarez-Gonzalez and Jacobson, 1987*), in competition with PARGs (*Pascal and Ellenberger, 2015*). In a third reaction, PARP1 uses $NAD^+$ to add ADPR to create branch points by generating linkages using the 2″-OH instead of the 2′-OH (*Figure 1A*). Branching has been shown to occur about once every 5–20 residues for autoPARylation (*Alvarez-Gonzalez and Jacobson, 1987*; *Miwa et al., 1979*) and is mechanistically feasible because of the pseudo-symmetry of ADPR that allows for binding of the elongating polymer in the opposite direction (*Ruf et al., 1998*). In the fourth and final reaction, PARP1 hydrolyzes $NAD^+$ to generate free ADPR. This 'treadmilling' reaction becomes more dominant at extended reaction times and low concentrations of $NAD^+$ (*Bauer et al., 1986*; *Desmarais et al., 1991*). We address the mechanism by which HPF1 affects initiation, elongation, and treadmilling in this article.

In our approach, we build on the seminal experiments performed by Ahel's group, who discovered HPF1 and validated its effect both in vitro and in vivo (*Bonfiglio et al., 2017a*; *Gibbs-Seymour et al., 2016*; *Suskiewicz et al., 2020b*). As we are particularly interested in studying how HPF1 mediates PARylation of histones, proteins that do not exist in free form in the cell, we use nucleosomes as both substrates and activators of PARP1. We have previously shown that nucleosomes bind tightly to PARP1 (1–13 nM) (*Clark et al., 2012*; *Muthurajan et al., 2014*). Of special relevance to both nucleosomes and HPF1, we have recently shown that HPF1 binds fivefold more tightly to a PARP1–nucleosome complex than to PARP1 bound to a DNA oligomer (*Rudolph et al., 2021*). We show that HPF1 not only changes the specificity of PARP1 from autoPARylation and glutamate modification towards transPARylation and serine modification, it also converts PARP1 into a strong $NAD^+$ hydrolase, resulting in much shorter PAR chains and high concentrations of free ADPR. Our results have important implications for the cellular role of HPF1 in PARylation, the PAR code, and the DNA damage response.

## Results

### Nucleosomes are better activators of PARP1 than free DNA

Since HPF1 is known to mediate PARylation of histones, proteins that are tethered to the genome in the form of chromatin, we set out to test the activity of PARP1 using nucleosome substrates. Understanding how HPF1 modulates the activity of PARP1 towards itself (autoPARylation) and towards histones (transPARylation) in the context of nucleosomes is complicated by the fact that nucleosomes provide both the DNA that activates PARP1 *and* serve as substrates for histone transPARylation. It is possible that different nucleosomes serve these two roles (*activator* vs. *substrate*) to different extents. Thus, we began our studies of the activity of PARP1 with nucleosomes by first investigating the ability of different nucleosomes to serve solely as *activators* of autoPARylation in the absence of HPF1 as under these conditions we do not observe transPARylation of histones (see below). We prepared a variety of nucleosome constructs, from mononucleosomes (Nuc147, Nuc165) to trinucleosomes (NLE-Tri, LE-Tri; *Figure 1C*), as well as tested free DNA in comparison (p18mer, p147mer, p165mer, p621mer). We found that all nucleosome complexes as well as free DNA bind to PARP1 with low nanomolar affinity as measured in a competition experiment using fluorescence polarization (FP; *Table 1*, $CC_{50}$ values).

AutoPARylation assays are made difficult by the fact that PARP1 is both the enzyme and its own substrate, thus limiting the range of enzyme concentrations for which one can detect activity (typically >30 nM) and precluding the use of many typical tools available to enzymological studies wherein enzyme and substrate concentrations is varied independently. We chose to measure autoPARylation activity by monitoring the incorporation of $^{32}P$-ADPR from $^{32}P$-$NAD^+$ onto PARP1 as this method does not rely on $NAD^+$-analogs, is highly sensitive, and can be implemented in modest throughput (*Figure 2A*). We were able to achieve reproducible and linear rates of autoPARylation using a variety of nucleosome and DNA activators (e.g., *Figure 2B*). As noted many years ago (*Alvarez-Gonzalez and Jacobson, 1987*; *Naegeli et al., 1989*; *Bauer et al., 1990*), we find that autoPARylation is a robust activity that leads to the addition of >100 ADPRs per PARP1 within 1 min at saturating concentrations of $NAD^+$. By varying the concentration of $NAD^+$, we determined that

**Table 1.** Comparison of the binding and activity of PARP1 using nucleosome and free DNA as activators, in the presence or absence of HPF1.

| PARylation product | PARP1 | | | | Histones |
|---|---|---|---|---|---|
| ± HPF1 | No HPF1 | No HPF1 | No HPF1 | With HPF1 | With HPF1 |
| DNA/nucleosome | $CC_{50}$ (nM) | $k_{cat}$ $(s^{-1})$ | ADPR (pmol/30 s) | ADPR (pmol/30 s) | ADPR (pmol/30 s) |
| p18mer | 11.5 ± 3.2 (n = 6) | 2.4 ± 0.8 (n = 14) | 6.6 ± 0.9 | 8.4 ± 3.8 | 5.4 ± 2.4 (peptide) |
| Nuc147 | 13.6 ± 6.4 (n = 4) | 4.0 ± 0.9 (n = 3) | 17.8 ± 2.1 | 6.7 ± 1.2 | 27 ± 5 |
| p165mer | 10.1 ± 4.3 (n = 5) | 2.7 ± 0.8 (n = 6) | n.d. | n.d. | n.d. |
| Nuc165 | 6.1 ± 2.7 (n = 8) | 4.4 ± 1.0 (n = 15) | 13.9 ± 2.0 | 5.2 ± 1.8 | 10.1 ± 1.3 |
| Nuc207 | n.d. | n.d. | 17.9 ± 4.5 | 8.1 ± 1.7 | 28 ± 12 |
| NLE-Tri | 1.6 ± 0.4 (n = 5) | 4.5 ± 1.2 (n = 4) | 20.3 ± 4.3 | 4.7 ± 1.3 | 15 ± 6 |
| p621mer | 4.2 ± 1.7 (n = 4) | 1.6 ± 0.4 (n = 2) | n.d. | n.d. | n.d. |
| LE-Tri | 1.9 ± 0.7 (n = 6) | 5.1 ± 1.3 (n = 3) | 18.5 ± 0.5 | 9.2 ± 0.2 | 29 ± 1 |

Column 1: The apparent competitive concentration for 50% binding ($CC_{50}$) values as determined by titrating nucleosomes or free DNA into a solution containing pre-bound PARP1 bound to fluorescent p18mer (p18mer*) and monitoring the release of p18mer* by fluorescence polarization. Column 2: The apparent $k_{cat}$ values (determined at $NAD^+$=40 µM = $K_m$ for $NAD^+$) were determined from fitting the measurements of activity for incorporation of ADPR as shown in **Figure 2B** in the plate-based assay at varying concentrations of DNA/nucleosome (**Figure 2D**). Data reflect at least three different replicates wherein each time point was collected in triplicate. Columns 3–6: incorporation of ADPR into PARP1, HPF1, and histones was determined using the gel-based assay using 10 µM $NAD^+$ after 30 s of reaction. Data shown are from 3 to 4 replicates for each assay condition. Activation by p18mer monitored addition of ADPR onto H3-tail peptide as there are no histones present in this reaction. All indicated errors are standard deviations from the mean. ADPR: ADP-ribose; PARP1: poly(ADP-ribose) polymerase 1; HPF1: Histone PARylation Factor 1.

the $K_m$ for $NAD^+$ is not affected by the activator (**Figure 2C**) and our $K_m$ values are in reasonable agreement with previous determinations of $K_m$ of $NAD^+$ for PARP1 (30–100 µM) using a variety of different assay methods (**Ruf et al., 1998**; **Desmarais et al., 1991**; **Miranda et al., 1995**; **Langelier et al., 2008**; **Jiang et al., 2010**). We noted that Nuc165 consistently yielded a higher turnover number ($k_{cat}$) than p18mer (p<0.0001), and these values are in reasonable agreement with previous determinations (**Ruf et al., 1998**; **Desmarais et al., 1991**; **Miranda et al., 1995**; **Langelier et al., 2008**; **Jiang et al., 2010**) of $k_{cat}$ for PARP1 (0.3–5 $s^{-1}$; **Table 1**). This difference in $k_{cat}$ was also determined for other nucleosome vs. free DNA comparisons (**Figure 2D**, **Table 1**). Note that p165mer and p621mer DNA fragments, used to assemble Nuc165 and LE-Tri nucleosomes, respectively, have $k_{cat}$ values that are similar to p18mer, whereas all the nucleosome complexes tested have similar, higher values of $k_{cat}$ (**Table 1**) despite having different lengths of overhanging DNA (**Figure 1C**). Direct comparison of these DNA fragments to their respective nucleosomes yields p values of 0.0024 for p165mer vs. Nuc165 and 0.0387 for 621mer vs. LE-Tri. We conclude that there is a unique aspect of the PARP1–nucleosome interaction compared to the PARP1-free DNA interaction that leads to more robust autoPARylation, and that this feature is independent of the type of nucleosome.

## HPF1 redirects PARylation from PARP1 onto histones

We next turned towards gaining a better understanding of how HPF1 redirects PARylation from PARP1 onto histones, and from modification of Glu/Asp to Ser residues in the context of nucleosomes serving as both activators and substrates of PARP1. Using Nuc165 and adding HPF1 in the plate-based assay described above, we saw no significant change in the $K_m$ for $NAD^+$ or the $k_{cat}$ for incorporation of ADPR into protein compared to the absence of HPF1, although the linearity of the assay was significantly reduced compared to assays in the absence of HPF1 (**Figure 2C**, **Figure 2— figure supplement 1**; note the non-zero intercepts with linear fits from 0 to 100 s). The similar overall levels of incorporation of ADPR in the presence and absence of HPF1 are qualitatively consistent with previous in vitro assays (**Gibbs-Seymour et al., 2016**) and comparisons of PAR levels by western

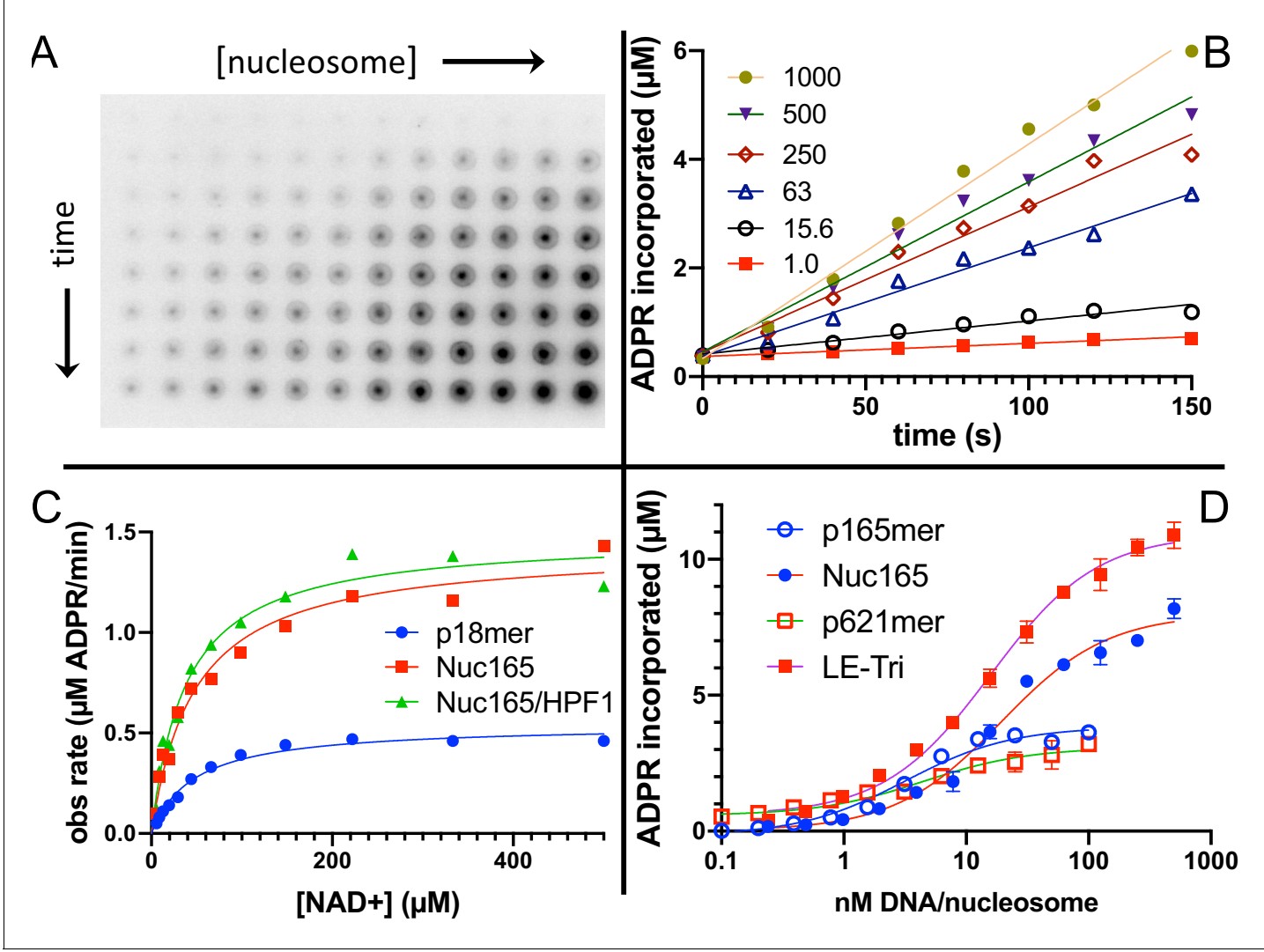

**Figure 2.** Nucleosomes are better activators of poly(ADP-ribose) polymerase 1 (PARP1) than free DNA. (**A**) Representative image from the filter binding assay monitoring the incorporation of $^{32}$P-NAD into PARP1 (autoPARylation) in the presence of Nuc165. Different time points (0, 20, 40, 60, 80, 100, 120, and 150 s) are represented in the vertical direction and different concentrations of Nuc165 (0.5–1000 nM by factors of 2 in concentration) are represented in the horizontal direction. (**B**) Representative data from monitoring the incorporation of $^{32}$P-NAD into PARP1 (autoPARylation) in the presence of Nuc165 (different concentrations indicated in nM). Good linearity of rates is observed at all concentrations of nucleosome up to 150 s. (**C**) Representative curve determining the K$_m$ for NAD$^+$ in the presence of p18mer, Nuc165, or Nuc165 in the presence of HPF1. The K$_m$ values for p18mer, Nuc165, and Nuc165 in the presence of HPF1 are 38 ± 9 µM (n = 8), 39 ± 10 µM (n = 3), and 54 ± 8 µM (n = 3), respectively. The k$_{cat}$ values for p18mer, Nuc165, and Nuc165 in the presence of HPF1 are 4.4 ± 1.9 s$^{-1}$, 2.4 ± 0.8 s$^{-1}$, and 4.6 ± 1.0 s$^{-1}$, respectively. (**D**) Representative activation curves for p165mer, Nuc165, p621mer, and LE-Tri demonstrating that nucleosomes lead to greater maximal autoPARylation activity. Indicated error bars are from triplicate assay points. Derived apparent values for k$_{cat}$ for these and other activators of PARP1 and their replicates are shown in *Table 1*.
The online version of this article includes the following source data and figure supplement(s) for figure 2:

**Source data 1.** Nucleosomes are better activators of PARP1 than free DNA.
**Figure supplement 1.** Addition of Histone PARylation Factor 1 reduces the linearity of poly(ADP-ribose) polymerase 1 (PARP1 )in a plate-based assay monitoring incorporation of ADP-ribose at short times of reaction.
**Figure supplement 1—source data 1.** Addition of HPF1 reduces the linearity of PARP1 in a plate-based assay monitoring incorporation of ADPR at short times of reaction.

blots in wild-type vs. HPF1-/- cells seen previously (*Palazzo et al., 2018*). However, in these reactions, PARylation occurs on PARP1, histones, and HPF1, and this plate-based precipitation assay does not allow for discrimination between the different reaction products. Given the high affinity (low nM) of nucleosomes for PARP1 (*Muthurajan et al., 2014*), it was not possible to separate

different targets of PARylation without denaturation. We thus used SDS-PAGE to analyze auto- vs. transPARylation. One complication of this method is the migration of $^{32}$P-NAD$^+$ at ~30 kDa in SDS-PAGE (*Figure 3A*). Thus, high concentrations of NAD$^+$ as one might use in a typical kinetic study cannot be used since the large signal from unused substrate obliterates the much fainter signal for measuring initial rates of PARylation for both PARP1 and histones. In fact, we note that the Ahel group typically performs their radioactive PARylation experiments at 5–100 µM $^{32}$P-NAD$^+$ for 20 min, thus ensuring the complete consumption of the potentially interfering substrate (*Bonfiglio et al., 2017a*; *Suskiewicz et al., 2020b*; *Fontana et al., 2017*; *Bartlett et al., 2018*) (see also Figure 5B). Alternatively, these types of PARylation assays have been successfully performed

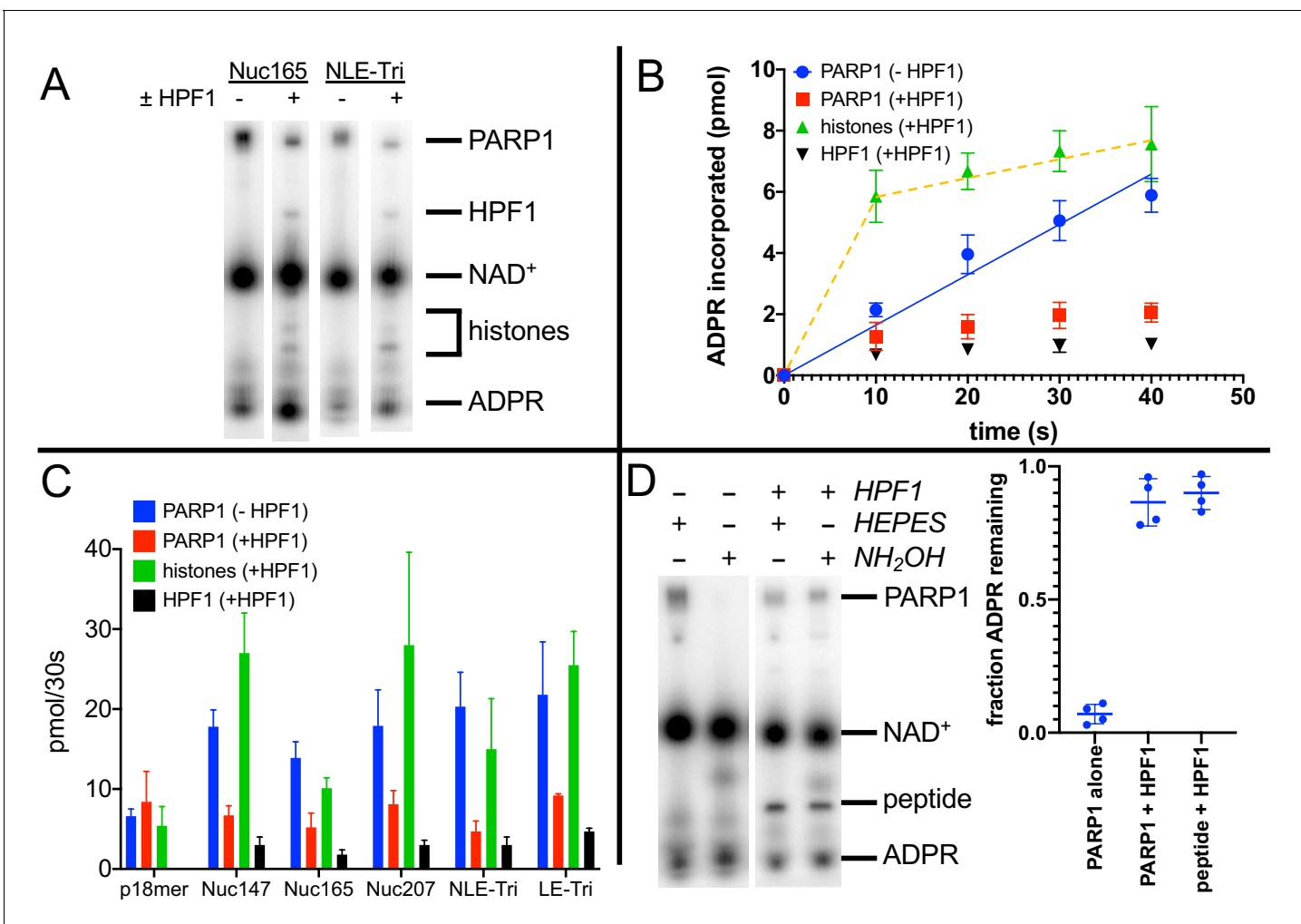

**Figure 3.** Histone PARylation Factor 1 (HPF1) mediates transPARylation onto nucleosomes and serines. (**A**) Representative autoradiogram from an assay gel demonstrating the reduction of autoPARylation in the presence of HPF1 with the concomitant appearance of PARylated histones and HPF1. The remaining substrate $^{32}$P-NAD$^+$ is by far the most prominent band on the image, indicating that we are monitoring the early time points of the reaction. (**B**) Time dependence of ADP-ribose (ADPR) incorporation onto poly(ADP-ribose) polymerase 1 (PARP1), histones, and HPF1. Note the linearity seen for autoPARylation in the absence of HPF1 and the non-linearity seen for PARylation in the presence of HPF1, as emphasized by the dotted line. (**C**) Bar graph of ADPR incorporation onto PARP1, histones, and HPF1 at 30 s demonstrating that in the presence of HPF1 histones become the primary target of PARylation to the detriment of PARP1. Levels of HPF1 PARylation are low despite the high concentration of HPF1 (2 µM) compared to nucleosomes (300–700 nM) in the reaction mixture. For the reaction indicated with p18mer, activation is by free DNA and the PARylated product is the H3 peptide. Each experiment was performed four separate times, and the data shown are mean values with standard deviations. (**D**) Representative gel demonstrating that PARylation in the absence of HPF1 is directed towards hydroxylamine labile Asp/Glu residues and in the presence of HPF1 becomes stable to this treatment, consistent with PARylation of Ser residues. Quantitation in the bar graph is the summary of 8–12 replicates. The online version of this article includes the following source data for figure 3:

**Source data 1.** HPF1 mediates transPARylation onto nucleosomes and serines.

using antibodies to detect modified proteins following separation on SDS-PAGE (*Bonfiglio et al., 2017a*; *Palazzo et al., 2018*; *Liszczak et al., 2018*), which avoids the problem of detecting NAD$^+$ but is generally not a good quantitative method. Thus, to better compare the relative activities of PARylation between PARP1 and nucleosomes, we used multiple turnover conditions with 10 μM $^{32}$P-NAD$^+$ at short time points (30 s). This method allows for simultaneous detection of PARylation of PARP1, HPF1, and histones (*Figure 3A*). These gels confirm that HPF1 reduces autoPARylation and mediates transPARylation of histones and HPF1. It is important to note that we do not detect any PARylation of histones (or H3 peptide) in the absence of HPF1.

To gain quantitative insights, we performed each of these experiments four times to derive values for the amount of ADPR incorporated into the various products, PARP1, histones (or H3 peptide), and HPF1. Validating our approach, controls in the absence of HPF1 monitoring autoPARylation yielded reasonably linear incorporation of ADPR vs. time, similar to what we observed using the plate-based assay (*Figure 3B*, blue circles). Additionally, this gel-based assay compares well with the plate assay when comparing levels of ADPR added to PARP1 at 30 s in that we see more total incorporation of ADPR in the presence of nucleosomes than free DNA (in the absence of HPF1) (*Table 1*, column 2, compare p18mer with all nucleosome constructs). We observe that for autoPARylation, in the absence of HPF1 and at these low concentrations of NAD$^+$, 3–7 ADPRs are attached per PARP1 within 30 s, consistent with good enzymological practices wherein one should consume low amounts of substrate (<10% of the initial NAD$^+$).

Strikingly, in the presence of HPF1 and nucleosomes, incorporation of ADPR onto PARP1, HPF1, and, most prominently, histones becomes less linear as if a faster 'burst of activity' had occurred followed by no further incorporation of ADPR (*Figure 3B*). This non-linearity correlates in part with the reduced linearity seen in the plate-based assay in the presence of HPF1 (*Figure 2C*, *Figure 2—figure supplement 1*) but may also be due in part to slower quenching kinetics by the use of olaparib instead of acid. This non-linearity precluded comparisons of rates of PARylation, and thus we compared the amounts of ADPR incorporation at 30 s. Because of the 'burst' of incorporation of ADPR onto histones in the presence of HPF1, this comparison at 30 s serves as a lower limit for the HPF1-mediated switch in specificity. AutoPARylation is consistently decreased by factors of 2–4 by the addition of HPF1, regardless of which nucleosome is used as an activator (*Figure 3B, C*, *Table 1*). The presence of HPF1 clearly makes histones in the nucleosome substrates the dominant recipient of PARylation (*Figure 3B, C*, *Table 1*). Despite the limitations of this assays, we conclude that the switch in specificity from autoPARylation towards transPARylation is minimally two- to fourfold (*Table 1*). Neither the HPF1-dependent decrease in autoPARylation nor the large switch in specificity is observed using p18mer DNA and H3 tail peptide as a substrate, presumably because of the weaker interaction of HPF1 with the PARP1–p18mer complex vs. the PARP1–Nuc165 complex (*Rudolph et al., 2021*). This result emphasizes the importance of studying HPF1 with nucleosomal substrates and activators. Additionally, we see a very small amount (<0.1 equivalents) of incorporation of ADPR onto HPF1 in the presence of nucleosomes but not free DNA (*Figure 3B*).

In addition to the HPF1-mediated switch in specificity from autoPARylation to histone PARylation, we also confirmed the concomitant switch in specificity from Glu/Asp modification to Ser modification. PARylation on Glu/Asp residues can be removed by treatment with hydroxylamine (HA) (*Cervantes-Laurean et al., 1997*). Prior to loading reaction mixtures with $^{32}$P-NAD$^+$ onto gels for quantitation, samples were either treated with 1 M HA (pH 7.0) or as a control with 1 M HEPES (pH 7.0). Treatment with HA removes >90% of ADPR from PARP1 in the absence of HPF1, as seen in a representative gel and by quantitation from replicate experiments (*Figure 3D*). In the presence of HPF1, PARylation on both PARP1 and H3 tail peptide is retained (>90%) after exposure to HA (*Figure 3D*), indicating the linkages to ADPR are no longer via Asp/Glu and most likely to Ser, as previously reported (*Bonfiglio et al., 2017a*).

We conclude that HPF1 significantly suppresses autoPARylation and thus makes histones in the nucleosome, and not PARP1, the primary target of PARylation. This effect is unique to nucleosome substrates as we do not observe it using activation by p18mer and the histone H3-derived peptide as a substrate. In addition to the switch in protein specificity from PARP1 to histones, HPF1 also mediates the dramatic switch in specificity from Glu/Asp residues to Ser residues.

## Glu284 is the catalytic residue that mediates transPARylation of histones

The discovery that PARylation in response to DNA damage in cells (*Larsen et al., 2018*; *Palazzo et al., 2018*) and in vitro (see our results above and *Bonfiglio et al., 2017a*) occurs primarily on serine residues instead of glutamate or aspartate requires a fundamental rethinking of the reaction mechanism of PARP1. Glutamate and aspartate residues are deprotonated under physiological conditions and thus primed for nucleophilic attack on the C1' of NAD$^+$ (*Figure 4A*). However, serine as a target for PARylation is protonated, and thus one might expect that deprotonation of serine by a catalytic base would be required to initiate the nucleophilic attack (*Figure 4A*). Deprotonation of serine to increase nucleophilicity is a familiar theme in enzymatic catalysis as, for example, in the

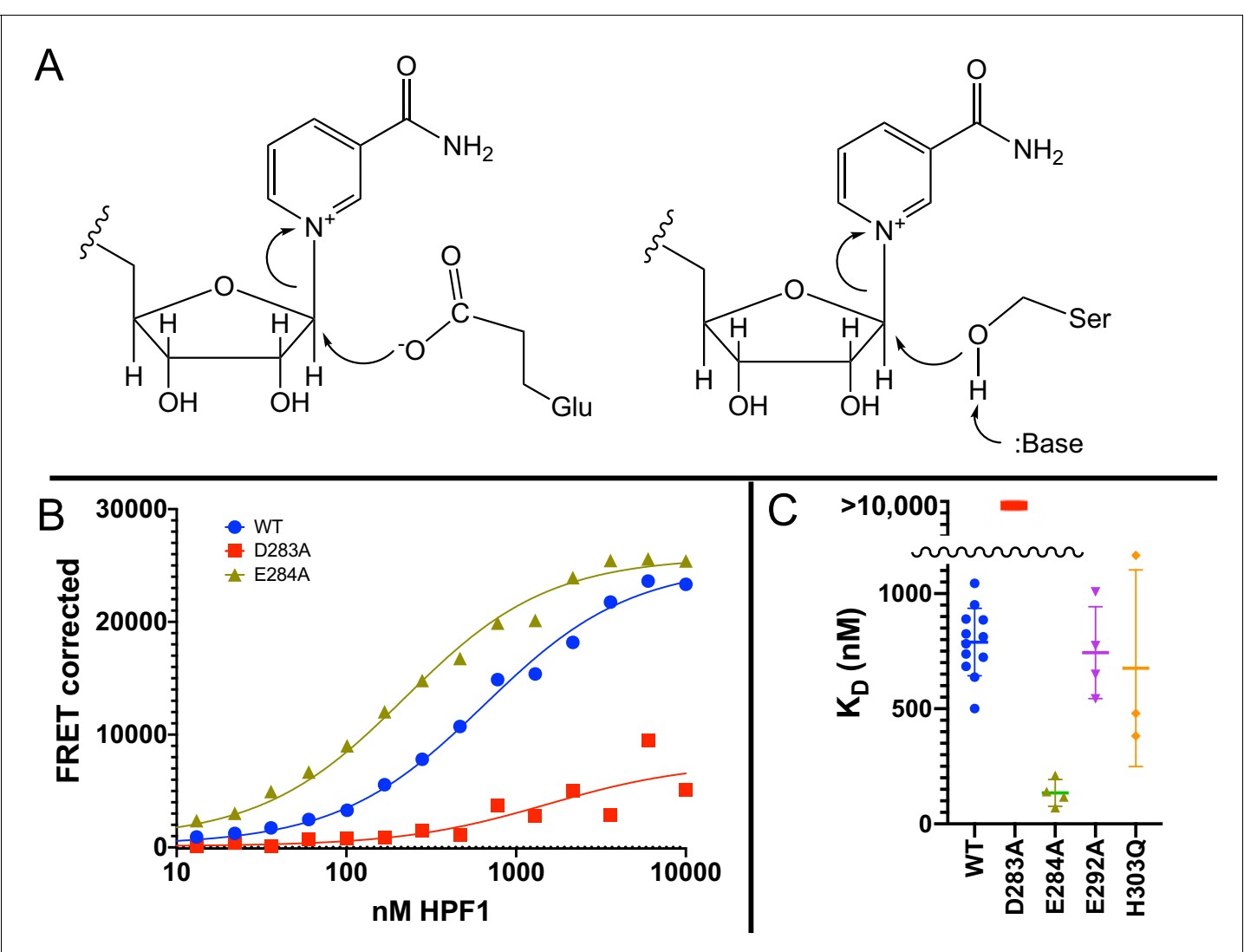

**Figure 4.** Glu284 of Histone PARylation Factor 1 (HPF1) is the catalytic base required for transPARylation. (**A**) Chemical mechanism of PARylation of glutamate (Glu, on left) does not require a catalytic base, whereas PARylation of serine (Ser, on right) requires deprotonation of serine by a catalytic base. (**B**) Representative curves demonstrating the binding of HPF1 (WT, D283A, and E284A) to the poly(ADP-ribose) polymerase 1 (PARP1)–Nuc165 complex using FRET between labeled HPF1 and labeled PARP1. (**C**) Bar graph for binding of HPF1 to PARP1–Nuc165 complex as determined by FRET assay demonstrating that the E284A mutant of HPF1 binds more tightly than WT, and that the D283 mutant does not bind with measurable affinity. The E292A and H303Q mutants of HPF1 are shown to bind with similar affinity as WT HPF1. Data for these findings with standard deviations and number of replicates can be found in *Table 2*.

The online version of this article includes the following source data for figure 4:

**Source data 1.** Glu284 of HPF1 is the catalytic base required for transPARylation.

serine proteases, esterases, and lipases (Ser–His–Asp catalytic triad) (*Dodson and Wlodawer, 1998*). Although PARP1 has a known catalytic base (Glu988) (*Marsischky et al., 1995*), this residue alone is not capable of mediating PARylation of serines as we see no histone PARylation without the addition of HPF1. With the discovery that HPF1 is responsible for the specificity switch from glutamate to serine residues, as well as mediating transPARylation (*Bonfiglio et al., 2017a*), we conjectured that HPF1 could provide the essential catalytic base. Given that we had no structural information for HPF1 at the time as our efforts began well before the publication of the crystal structure of HPF1 bound to PARP2 (*Suskiewicz et al., 2020b*), we set out to find the putative catalytic base by mutagenesis of likely candidate residues that were either histidines, aspartates, or glutamates. Our criterion for success was to identify a residue whose mutation abolished transPARylation activity without being detrimental to the stability of the protein or its binding to a PARP1–Nuc165 complex. Site-directed mutagenesis looking for loss of catalytic function with retention of structure is a classical approach in enzymology that has been used to successfully identify many catalytic residues prior to the determination of three-dimensional structures (*Plapp, 1995*). We measured activity using gel-based PARylation assays as in *Figure 3* and stability of the protein using thermal denaturation experiments. We previously developed a FRET-based assay for quantitating the binding interaction between HPF1 and PARP1, demonstrating that HPF1 prefers the PARP1–Nuc165 complex vs. a PARP1–p18mer complex by a factor of five (790 nM vs. 3800 nM) (*Rudolph et al., 2021*).

To select potential catalytic base residues, we first aligned the sequences of all known HPF1-like proteins using TF317026 from treefam.org as a source for 101 different family members. The family tree consists of two major branches, and we noted that the two residues in the C-terminal domain of HPF1 (Tyr238 and Arg239) previously identified as important for interaction with PARP1 by the Ahel group (*Gibbs-Seymour et al., 2016*) were conserved in both branches. Given that there are 62 histidines, aspartates, and glutamates in HPF1, we assumed that a catalytic base would be highly conserved, and therefore considered only completely conserved residues in the C-terminal domain that had been shown to be critical for the interaction with PARP1. This pruning procedure leaves only Asp283, Glu284, Asp286, Glu292, Asp296, and His303, which were individually mutated to alanine.

We performed transPARylation assays with the mutant proteins as described above, demonstrating that the D283A and E284A mutants of HPF1 were unable to mediate histone or HPF1 PARylation, in contrast to WT HPF1 or other mutations such as the E292A and His303Q mutations (*Table 2*). All four HPF1 mutants displayed similar protein stability in denaturation assays (*Table 2*), suggesting that the loss of transPARylation for D283A and E284A was not due to misfolding. However, only three of these mutants retained their binding to the PARP1–Nuc165 complex (*Figure 4C, D*), with the D283A mutant of HPF1 showing no detectable formation of a complex ($K_D$ >10,000 nM). Interestingly, the E284A mutant bound significantly tighter (p<0.0001) to the

**Table 2.** HPF1 mutant analysis.

| HPF1 | Melting temperature (°C) | Histone PARylation (pmol ADPR/30 s) | $K_D$ (nM) |
|---|---|---|---|
| WT | 50.6 ± 0.10 n = 5 | 10.1 ± 1.3 n = 4 | 790 ± 147 n = 12 |
| D283A | 50.8 ± 0.11 n = 5 | 0 | >10,000,000 n = 5 |
| E284A | 50.3 ± 0.86 n = 5 | 0 | 135 ± 58 n = 4 |
| E292A | 49.0 ± 0.08 n = 5 | 17.5 ± 2.1 n = 4 | 743 ± 200 n = 4 |
| H303Q | 50.2 ± 0.10 n = 5 | 21.1 ± 7.6 n = 4 | 676 ± 427 n = 3 |

Column 1: protein stability was measured by Thermo Fisher Protein Thermal Shift kit. Column 2: incorporation of ADPR onto histones was determined using the gel-based assay using 10 µM NAD$^+$ after 30 s of reaction. Data shown are from 3 to 4 replicates for each assay condition. Column 3: binding constant of HPF1 to the PARP1–Nuc165 complex was determined using the FRET assay shown in *Figure 4B*. Data shown are the mean and standard deviation of the indicated number of replicates. HPF1: Histone PARylation Factor 1; PARP1: poly(ADP-ribose) polymerase 1; ADPR: ADP-ribose.

PARP1–Nuc165 complex than wild-type HPF1 (135 vs. 790 nM; *Figure 4D*, *Table 2*), validating its structural integrity and ability to recognize the PARP1–Nuc165 complex despite not promoting transPARylation activity. The loss of transPARylation and the retention of binding to the PARP1–Nuc165 complex indicated that Glu284 is the catalytic base required for PARylation of serine residues (*Figure 4A*), whereas Asp283 is required for a productive interaction of HPF1 with the PARP1–Nuc165 complex. Our findings are an independent validation of the identification of Glu284 as the catalytic base from recent structural and biochemical analyses by the Ahel group (*Suskiewicz et al., 2020b*). Additionally, the discovery of the E284A mutant allowed us to further dissect how HPF1 modulates the activity of PARP1, namely (1) catalytic, by providing a base for deprotonation of Ser residues (*Figure 4A*), and (2) binding, by perturbing the shared active site formed by direct interactions with the PARP1–Nuc165 complex.

## HPF1 (WT and E284A) converts PARP1 into an NAD$^+$ hydrolase and yields much shorter PAR chains

Although gel-based PARylation assays provide insight into the addition of ADPR onto histones (in the context of nucleosomes) vs. PARP1 at short extents of reaction, they do not address the outcome of PARylation in terms of chain length. It is important to address the effect of HPF1 on the final outcome of PARylation (i.e., short vs. long PAR chains) because the different DNA repair factors may be recruited to different modifications (i.e., PAR code; *Aberle et al., 2020*; *Karlberg et al., 2013*). We therefore developed two novel  assays using high pressure liquid chromatography (HPLC). The first assay simultaneously quantitates ADPR, NAD$^+$, and nicotinamide with high accuracy and sensitivity (*Figure 5A*), which allows us to monitor overall consumption of NAD$^+$ and whether ADPR equivalents are attached to protein or released as free ADPR (see *Figure 1B*). All assays are linear with respect to nicotinamide formation for at least 90 s (*Figure 5A*, inset), and we chose to compare PARP1 activity at 60 s under different reaction conditions (i.e., ±HPF1). The second assay again monitors consumption of NAD$^+$ and formation of ADPR and nicotinamide, this time for the full extent of the reaction to where all the NAD$^+$ has been consumed, and then subsequently analyzes chain extension via the quantitation of AMP-PR, which reflects the 'middle' pieces of PAR chains after digestion by phosphodiesterase (*Figure 6A*). Using these assays, again with Nuc165 as both activator and substrate in comparison with oligomeric DNA (and autoPARylation), we made a number of surprising discoveries while also confirming our results described above.

First, in agreement with both the plate-based and gel-based radioactive assays described above, Nuc165 (in the absence of HPF1) yields approximately twofold higher turnover of NAD$^+$ and formation of nicotinamide than p18mer (12.5 vs. 5.4 µM in 1 min; *Figure 5B*). Addition of HPF1 increases the overall turnover of NAD$^+$ to form free nicotinamide by a factor of four in the presence of Nuc165 (47 vs. 12.5 µM in 1 min; *Figure 5B*). The effect of HPF1 is not as pronounced when using p18mer instead of Nuc165 (*Figure 5B*), presumably because of its weaker interaction with the PARP1–p18mer complex compared to the PARP1–Nuc165 complex (*Rudolph et al., 2021*). Interestingly, this HPF1-dependent effect (using Nuc165 or p18mer) is at least partially conserved for the E284A mutant of HPF1 (*Figure 5B*), which binds to the PARP1–Nuc165 complex (*Figure 4C, D*), but does not promote transPARylation (*Table 2*). Very dramatically, the modest HPF1-dependent increase in the formation of nicotinamide is eclipsed by a much larger increase in ADPR formation (*Figure 5C*). In the absence of HPF1, we observe 10–14% 'treadmilling', while in the presence of HPF1, PARP1 spends most of its catalytic power (~90%) consuming NAD$^+$ to form free ADPR. Surprisingly, the E284A mutant of HPF1, which binds more tightly than wild-type HPF1, has the same effect on the hydrolase activity of PARP1, even though it neither promotes transPARylation nor serine PARylation (*Figure 5C*). Control experiments demonstrated that neither wild-type HPF1 nor the E284A mutant have detectable levels of NAD$^+$ hydrolase activity on their own. The observed dramatic increase in treadmilling explains the lack of linearity (i.e., the burst in PARylation) seen in the gel-based assays with HPF1 above (*Figure 3B*). That is, although the initiation reaction on serines occurs efficiently, PARP1 then spends most of its catalytic power performing hydrolysis of NAD$^+$, thereby precluding further attachment of ADPR onto protein in the presence of HPF1. These unexpected results suggest that HPF1 converts PARP1 into a strong NAD$^+$ hydrolase and presumably represses formation of long PAR chains on histones.

For PARP1 alone, autoPARylation is rapid and efficient wherein the end of the growing PAR chain is located in a binding pocket adjacent to the primary NAD$^+$ binding site (*Ruf et al., 1998*). In the

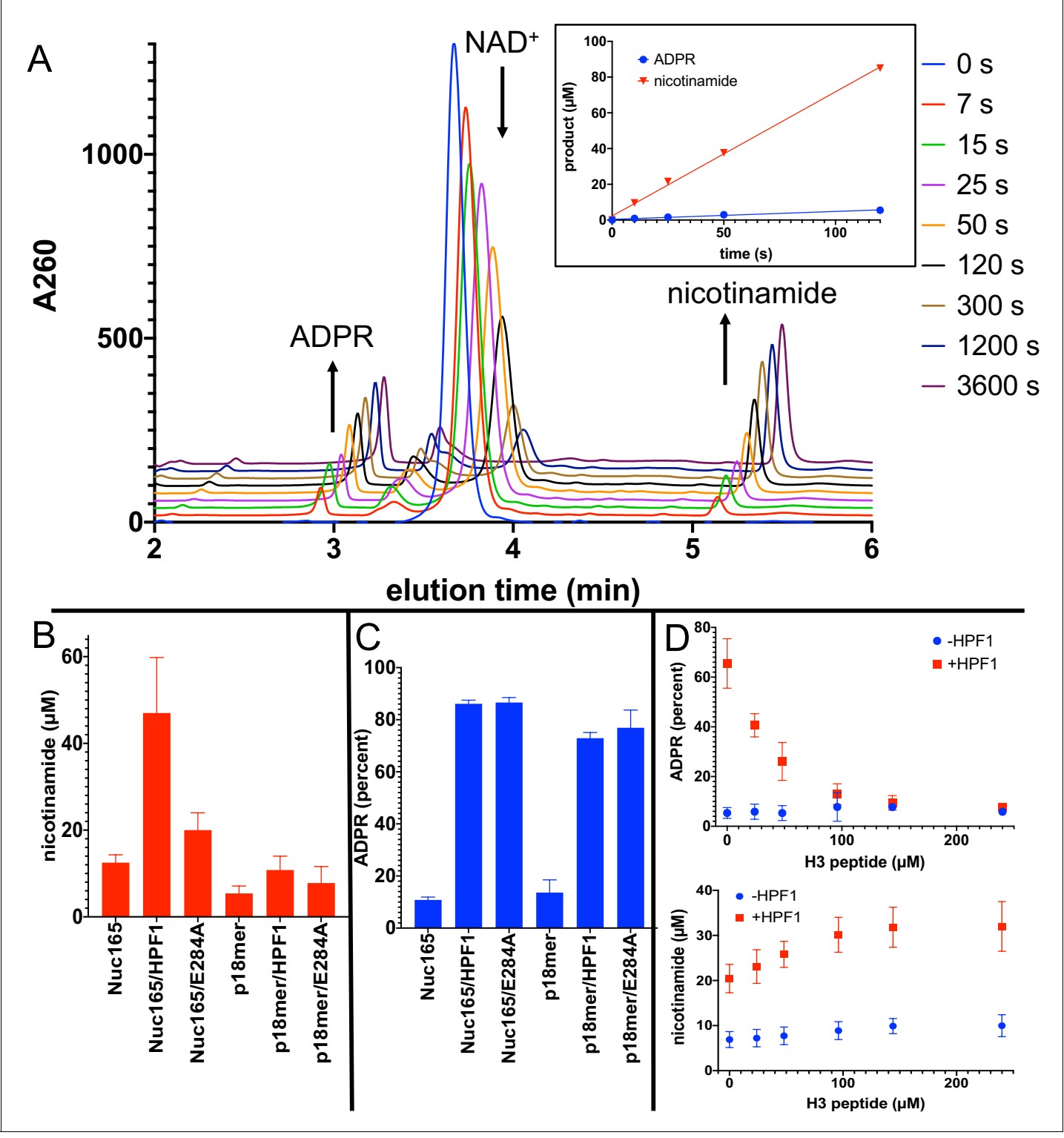

**Figure 5.** Addition of Histone PARylation Factor 1 (HPF1) converts poly(ADP-ribose) polymerase 1 (PARP1) predominantly into an NAD+ hydrolyase. (**A**) Representative HPLC traces from a reaction of PARP1 using p18 as an activator that simultaneously monitors depletion of NAD+ (200 µM initial) and formation of both ADP-ribose (ADPR) and nicotinamide. Each successive time point trace is offset in both the x- and y-axis to allow for better visualization. The inset shows that the assay is linear with respect to formation of nicotinamide and ADPR for at least 90 s. (**B**) Comparison of activity of PARP1 (100 nM) as measured by formation of nicotinamide using either Nuc165 (300 nM) or p18mer (100 nM) as activators in the presence or absence of HPF1 (2 µM, wild-type vs. E284A mutant) after 1 min of reaction time. (**C**) Comparison of percent of turnover of PARP1 that leads to free ADPR under

*Figure 5 continued on next page*

*Figure 5 continued*

the same conditions as in (B). Error bars in (C) and (D) are derived from three experiments, each performed using four replicates. (D) The hydrolase activity of PARP1 (100 nM) is suppressed (top panel) with a modest increase in overall activity as measured by nicotinamide formation (bottom panel) by high concentrations of H3 histone tail peptide (varied as indicated), but only in the presence of HPF1 (2 µM). Reactions were performed at 200 µM NAD$^+$ for 1 min, and the data and standard deviations shown are derived from four separate experiments.

The online version of this article includes the following source data for figure 5:

**Source data 1.** Addition of HPF1 converts PARP1 predominantly into an NAD$^+$ hydrolase.

recent structure of HPF1 bound to PARP2, HPF1 partially occupies the binding site typically associated with PAR chain extension (*Suskiewicz et al., 2020b*). To test whether the increased hydrolase activity is due to the lack of readily available substrate for PARylation due to HPF1 blockage of this extension site, we performed a titration of the histone H3 tail peptide in the presence of a fixed concentration of p18mer as a DNA activator. Providing high concentrations of the substrate peptide suppressed the treadmilling effect to baseline levels in the presence of HPF1 while subtly increasing the overall activity of PARP1 (*Figure 5D*). Control experiments in the absence of HPF1 showed no change in the percentage of hydrolysis or total turnover as a function of H3 tail peptide (*Figure 5D*).

To more directly address the profound effect of HPF1 on the length of PAR chains, we modified our assay conditions by using greater amounts of protein to facilitate quantitation of AMP-PR, the product formed after digestion by phosphodiesterase that represents chain 'middles' (*Figure 6A*). Monitoring the entire time course of NAD$^+$ disappearance and ADPR formation (*Figure 6B*), we observe faster depletion of NAD$^+$ and much greater formation of ADPR in the presence of HPF1, in agreement with the fixed time point results in *Figure 5C, D*. If we compare the incorporation of AMP-PR ('middles') at the end of the reaction when all of the NAD$^+$ (200 µM initially) has been consumed, we confirm that the PAR chains are much shorter in the presence of HPF1 than in its absence. This holds true for activation by both nucleosomes and p18mer but is more pronounced for nucleosomes, which we again attribute to the higher affinity of HPF1 for the PARP1–Nuc165 complex compared to the PARP–p18mer complex. Control experiments demonstrated that HPF1 or the E284A mutant alone did not exhibit significant phosphodiesterase activity. Given the results above, we were not surprised to find that the E284A mutant of HPF1 had a similar effect as wild-type HPF1 on PAR chain formation. To be noted, HPF1-mediated PARylation is not limited to addition of a single ADPR (MARylation) as we do observe significant concentrations of AMP-PR that can arise only through true polymerization (minimally trimers; *Figure 6D*).

## Discussion

We began our work by demonstrating that nucleosomes are better activators of PARP1 than free DNA (*Figure 2*, *Table 1*). Because the tested nucleosomes all activate equally well (regardless of overhang length, or mono- vs. tri-nucleosomes), our results extend previous observations about the promiscuity of PARP1 with respect to DNA activators (*Langelier et al., 2014*; *D'Silva et al., 1999*; *Liu et al., 2017*) using a more physiologically relevant activator. Based on the seminal work by Pascal and Black demonstrating that DNA activation of PARP1 is mediated by a series of conformational changes that disorder the HD domain and thereby open up the active site for binding of NAD$^+$ (*Dawicki-McKenna et al., 2015*; *Steffen et al., 2016*), we hypothesize that nucleosomes are in some way more efficient in this process. Perhaps there are direct interactions between the unfolded (or alternatively folded) HD domain and nucleosomes that stabilize the open conformation and prevent this domain from collapsing back onto the catalytic domain. In fact, just such an interaction is observed by the loop in the HD domain of PARP2 (Leu302 and Arg303) with the linker DNA from the nucleosome (*Bilokapic et al., 2020*). Ongoing structural efforts will hopefully show whether this or some other interaction exists for PARP1, leading to the enhanced activation of PARP1 by nucleosomes vs. free DNA.

Upon adding HPF1 to assays using nucleosomes as both activators and substrates of PARP1, we show that the activity of PARP1 is significantly redirected towards histones and away from PARP1 (*Figure 3*, *Table 1*). Our results are consistent with those from the Ahel group obtained with free DNA and histone-derived peptides (*Gibbs-Seymour et al., 2016*) as well as from the Muir group using nucleosomes (*Liszczak et al., 2018*), but provide quantitative information regarding the extent

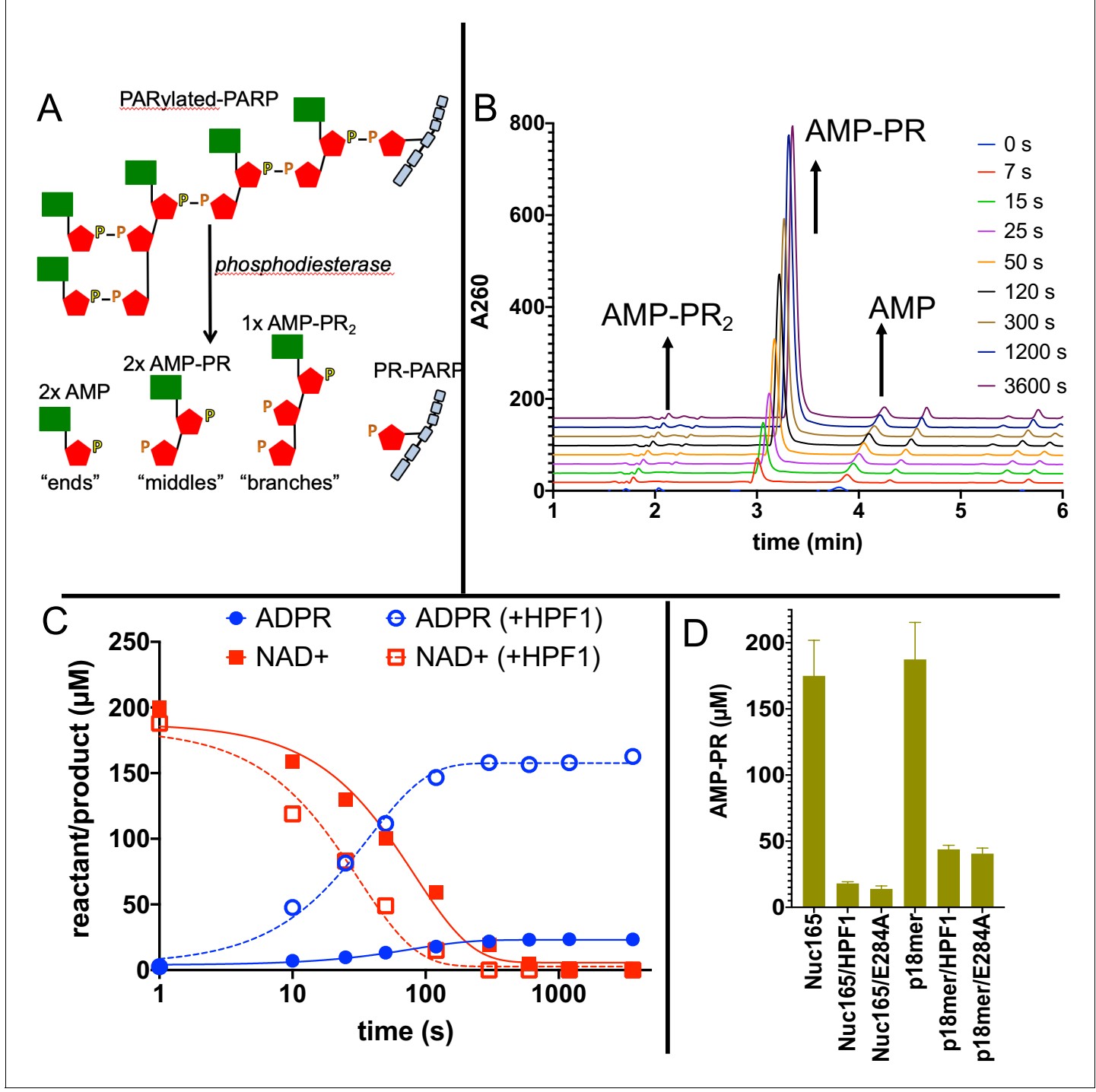

**Figure 6.** Addition of Histone PARylation Factor 1 (HPF1) accelerates consumption of NAD[+] and leads to significantly shorter PAR chains. (**A**) Cartoon representation of the reaction products generated by phosphodiesterase treatment of PARylated poly(ADP-ribose) polymerase 1 (PARP1). (**B**) Representative HPLC traces from the analysis of PARylated PARP1 that used p18 as an activator, simultaneously monitoring AMP ('ends'), AMP-PR ('middles'), and AMPR-PR$_2$ ('branches'). Each successive time point trace is offset in both the x- and y-axis to allow for better visualization. (**C**) Comparison of a time course for accumulation of ADP-ribose (ADPR) and depletion of NAD[+] for PARP1 with Nuc165 in the absence (solid symbols and lines) or presence (open symbols, dotted lines) of HPF1 (wild-type). Note the faster consumption of NAD[+] in the presence of HPF1 and the much higher levels of ADPR observed. The appearance of nicotinamide (not shown for clarity) was also monitored in these assays and mirrored the depletion of NAD[+]. (**D**) Quantitation of AMP-PR (chain middles) released from PARylated PARP1 and histones after complete consumption of 200 μM NAD[+]. Note that in the absence of HPF1 ~90% of the ADPR is attached to protein, indicating long PAR chains. In contrast, the addition of HPF1 (either wild-type or

*Figure 6 continued on next page*

*Figure 6 continued*

E284A mutant) led to much shorter PAR chains, consistent with the high amount of free ADPR formed (*Figures 5D,* C). Error bars in (D) are derived from three experiments.

The online version of this article includes the following source data for figure 6:

**Source data 1.** Addition of HPF1 accelerates consumption of NAD$^+$ and leads to significantly shorter PAR chains.

of this switch. For example, HPF1 causes transPARylation of histones to be preferred over autoPARylation of PARP1 minimally by factors of 2–4 (*Figure 3C*, *Table 2*), a switch that is particularly dramatic as we observe no histone PARylation in the absence of HPF1 (*Figure 3A*). This switch in specificity is notable as autoPARylation is a simple one-body problem wherein PARP1 modifies itself, whereas transPARylation requires the correct interactions between PARP1, HPF1, and the histone tails in the nucleosomes. Also, it should be re-emphasized that, in addition to the switch towards transPARylation, HPF1 mediates the switch in amino acid specificity away from aspartate/glutamate towards serine, even on PARP1 itself (*Figure 3D*).

In an independent and unbiased approach to better understand the mechanism of how HPF1 effects the switch in amino acid specificity from glutamates and aspartates to serines, we have identified Glu284 as the catalytic base on HPF1 (*Figure 4*, *Table 2*), as also recently published by the Ahel (*Suskiewicz et al., 2020b*) and Yun groups (*Sun et al., 2021*). Importantly, we demonstrate that the E284A mutant has the same stability as wild-type HPF1 and binds even more tightly to the PARP1–nucleosome complex than wild-type HPF1. The occurrence of a shared active site formed by PARP1 and HPF1 is a completely unexpected and novel finding in the field of post-translational modification (*Suskiewicz et al., 2020a*). The PARP1–HPF1 complex has two catalytic bases. Glu284 on HPF1 mediates the deprotonation and activation of serine residues on acceptor proteins for the initiation reaction (*Figure 4A*), while Glu988 on PARP1 mediates the deprotonation and activation of the 2′- or 2″-OH on the acceptor ADPR for PAR chain extension or branching, respectively (*Figure 1A, B*). Glu988 on PARP1 also facilitates the binding of NAD$^+$ via an H-bond to the 3′-OH (*Ruf et al., 1998*). Consistent with Glu284 on HPF1, not Glu988 on PARP1, performing the de-protonation required for initiation, Ahel's group (*Fontana et al., 2017*) observe HPF1-mediated incorporation of ADPR by the E988K mutant of PARP1, albeit at lower levels than wild-type.

Our most dramatic and unexpected finding is the extent by which HPF1 converts PARP1 from a polymerase that adds long polymers of ADPR onto itself or histones, into a hydrolase that generates free ADPR. Under our conditions, almost 90% of all the NAD$^+$ consumed by PARP1 in the presence of nucleosome activators and HPF1 is converted to free ADPR (*Figure 5*). This observation holds true both at short time points of the reaction (*Figure 5D*) and at the end of the reaction where all the NAD$^+$ has been consumed (*Figure 6C*). Also, recall that the presence of both nucleosomes and HPF1 increases the overall turnover of NAD$^+$ by PARP1 (*Figure 5C*, *Figure 6C*). These dual effects lead to rapid depletion of NAD$^+$ and formation of high concentrations of free ADPR. Given that increasing PARylation acceptor sites (i.e., high concentrations of H3 histone tail peptide, *Figure 5D*) greatly reduces the percentage of free ADPR formed, we conclude that HPF1 blocks PAR chain formation by occluding the extension site on PARP1 (*Suskiewicz et al., 2020b*), and thus PARP1, still bound to activating DNA (i.e., nucleosome), quickly runs out of suitable PARylation sites and therefore uses water as a nucleophile instead. The affinity of PARP1 for nucleosomes is very high (nM), and it is thought that significant PARylation is required for dissociation of DNA from PARP1 (*Rudolph et al., 2021*; *Langelier et al., 2014*). Thus, it is unclear what ratio of treadmilling would occur under cellular conditions where rapid exchange of PARP1 to other nucleosomes is most likely not occurring, and new methods will be needed to investigate this quantitatively. The high consumption of NAD$^+$ by PARP1 suggests that PARP1 not only plays a role in the DNA damage response but also in metabolic homeostasis (*Hurtado-Bagès et al., 2020*; *Murata et al., 2019*). NAD$^+$ is electron acceptor in glycolysis and the citric acid cycle. PARP1 is the major consumer of NAD$^+$ in the nucleus, and the rapid depletion of the NAD$^+$ pool (50–80%) by PARP1 in response to DNA damage leads to ATP depletion, has dramatic consequences for mitochondria, and leads to cell death via apoptosis and necrosis (*Berger, 1985*). The observation of partitioning away from PARylation and towards ADPR formation by HPF1 is a novel finding that suggests a more important role for ADPR than anticipated as a signaling molecule in its own right (*Perraud et al., 2001*; *Qi et al., 2019*).

The conversion of PARP1 into an NAD$^+$ hydrolase is also evident in the much lower incorporation of chain 'middles' onto histones or PARP1 (*Figure 6D*), indicating that HPF1-mediated PARylation yields much shorter chains than the typical long chains seen during autoPARylation in the absence of HPF1. Our quantitative demonstration of these shorter chains is in agreement with previous observations that smears of PARP1 typically seen after autoPARylation in SDS-PAGE are reduced in the presence of HPF1 (*Gibbs-Seymour et al., 2016*) and that PAR chains are much reduced in amount and length as detected by enzymatic labeling of terminal ADPR (*Ando et al., 2019*). Additionally, it is interesting to compare our findings with a recent demonstration that HPF1-mediated PARylation in cells leads primarily to mono-ADPR on serine residues, in part due to the activity of the glycohydrolase PARG (*Bonfiglio et al., 2020*), which interestingly also generates ADPR. PARG has also been shown to mediate transPARylation of histones, in part by competing with PARP1 for PAR chains (*Ménard et al., 1990*; *Thomassin et al., 1992*; *Lagueux et al., 1995*). It therefore remains to be elucidated whether the rapid and sustained depletion of NAD$^+$ after DNA damage in intact cells (*Juarez-Salinas et al., 1979*) is due to PARylation and subsequent dePARylation by PARG, PARylation, and futile treadmilling by PARP1 alone, or some combination of these events. As with the increased NAD$^+$ hydrolase activity triggered by HPF1, the formation of shorter PAR chains is partially preserved using the catalytic HPF1 mutant E284A. As concluded for the high hydrolase activity, these shorter PAR chains using either wild-type HPF1 or its E284A mutation are readily understandable given that HPF1 partially occupies the binding site typically associated with PAR chain extension (*Suskiewicz et al., 2020b*). Although our assay did not allow us to measure chain branches with sufficient sensitivity, we would argue that for this same reason HPF1-mediated PARylation has few if any branch points.

The formation of shorter PAR chains in the presence of HPF1 has important consequences that become particularly relevant when considering intracellular concentrations of these proteins. PARP1 is a very abundant nuclear protein, exceeding the concentration of HPF1 by about twentyfold (*Gibbs-Seymour et al., 2016*). Also, PARP1 is one of the very first proteins recruited to DNA damage sites induced by laser microirradiation, whereas HPF1 arrives more slowly (*Aleksandrov et al., 2018*; *Mahadevan et al., 2019*). Thus, it is possible that substantial PARylation at some points in time, at some sites, or at some types of damage occurs without HPF1. The consequences for the so-called PAR code (*Aberle et al., 2020*; *Karlberg et al., 2013*) are immense. PARP1-alone-mediated autoPARylation leads to long and branched chains, whereas HPF1-mediated PARylation leads to short, most likely unbranched chains. This change in the PARylation pattern may result in the recruitment of different repair proteins to DNA damage sites with HPF1 vs. those without, and thus in turn may direct DNA damage to be repaired by one pathway over another. The differentiation between these two outcomes in a cellular context is difficult but worthy of further investigation.

## Materials and methods

**Key resources table**

| Reagent type (species) or resource | Designation | Source or reference | Identifiers | Additional information |
|---|---|---|---|---|
| Peptide, recombinant protein | PARP1 | UniProt | P09874 | As described in *Rudolph et al., 2018* |
| Peptide, recombinant protein | HPF1 | UniProt | Q9NWY4 | As described in manuscript |
| Sequenced-based reagent | p18mer DNA | IDT | | 5'-phosphate-GGGTTGCGGCCGCTTGGG-3'; double-stranded |
| Strain, strain background (*Escherichia coli*) | Rosetta DE3 pLys | EMD Millipore | 70956 | Chemically competent cells |
| Other | 384-well plates | Corning | 3575 | |
| Other | 96-well plates | Corning | 3898 | |
| Other | Whatman GF/C glass paper | Whatman | 28497-619-PK | |
| Chemical compound, drug | Olaparib | SelleckChem | S1060 | |

*Continued on next page*

*Continued*

| Reagent type (species) or resource | Designation | Source or reference | Identifiers | Additional information |
|---|---|---|---|---|
| Commercial assay or kit | QuikChange II Mutagenesis kit | Agilent Technologies | 200523 | |
| Other | Synergi Fusion-RP column | Phenomenex | 00F-4424-EO | |
| Peptide, recombinant protein | Nuc147, Nuc165, Nuc207, NLE-Tri, LE-Tri | Prepared in-house | | See *Muthurajan et al., 2016* |
| Chemical compound, drug | ADP-ribose | Sigma-Aldrich | A0752 | |
| Chemical compound, drug | Nicotinamide | Sigma-Aldrich | N3376 | |
| Chemical compound, drug | $NAD^+$ | Sigma-Aldrich | N0632 | |
| Peptide, recombinant protein | H3 peptide | Anaspec | AS-61701 | |
| Chemical compound, drug | $^{32}P$-$NAD^+$ | PerkinElmer | NEG023X | |
| Chemical compound, drug | Snake venom phosphodiesterase | Worthington | LS003926 | |
| Software, algorithm | GraphPad Prism 9.0 | GraphPad Prism | Version 9.0.1 (128) | |

## Materials

$NAD^+$, nicotinamide, and ADPR were purchased from Sigma-Aldrich. Snake venom phosphodiesterase was purchased from Worthington. DNA oligonucleotides and their complementary strands were obtained from IDT: p18mer: 5′-phosphate-GGGTTGCGGCCGCTTGGG-3′. $H3^{1-21}$ peptide was purchased from Anaspec. $^{32}P$-$NAD^+$ was purchased from PerkinElmer. Different nucleosomes (*Figure 1A*) using the Widom601 sequence were prepared as previously described (*Muthurajan et al., 2016*). Wild-type PARP1 was expressed and purified as previously described (*Rudolph et al., 2018*).

## Site-directed mutagenesis, expression, and purification of HPF1

Site-directed mutants for HPF1 were generated using the QuikChange II kit from Agilent Technologies according to manufacturer's specifications. All mutations were verified by DNA sequencing. Partial purification of HPF1 for initial evaluation of activity was performed as follows. Plasmids were transformed into Rosetta DE3 pLys cells (EMD Millipore), and transformants were grown in 100 mL $2\times$ YT media with 50 μg/mL kanamycin at 37°C until the $OD_{600}$ was ~0.6. Induction of protein expression was induced by addition of 1 mM IPTG (Gold Biotech), and cells were grown overnight at 18°C. Cell pellets collected by centrifugation were lysed by sonication (4 min at 30% power, Branson) in buffer containing 500 mM NaCl, 50 mM Tris-HCl (pH 8.0), 20 mM imidazole, 10 mM β-mercaptoethanol, 1.5 mM $MgCl_2$, 5 μL/5 mL benzonase (Novagen), and 20 mg lysozyme (Sigma-Aldrich). Cleared lysate was added to 200 μL of Nickel beads (Gold Biotech) in a small drip column (BioRad), reapplied several times, and HPF1 was subsequently recovered using elution buffer (50 mM Tris-HCl [pH 8.0], 500 mM NaCl, 250 mM imidazole, and 10 mM β-mercaptoethanol). The partially purified protein was concentrated (10 kDa cut-off, EMD Millipore) and exchanged into 100 mM potassium phosphate buffer (pH 8.0) prior to flash freezing and storage at −80°C. Full purification of HPF1 to homogeneity was performed essentially as previously described (*Gibbs-Seymour et al., 2016*), with the slight modification of using a gradient for the nickel affinity column from 0% to 40% over 15 CV. The stability of HPF1 (WT and all mutants) was evaluated using the ThermoFisher Protein Thermal Shift kit according to manufacturer's instructions using a qPCR. Melting temperatures were determined by identifying the low point of the peak after taking the derivative of the data.

## Competition experiment for measuring binding of nucleosomes and free DNA to PARP1 using FP

Nucleosomes and free DNA (20–5000 nM) diluted in binding buffer (50 mM Tris-HCl [pH 8.0], 50 mM NaCl, 1 mM $MgCl_2$, 0.1 mM EDTA, and 0.01% IGEPAL) were titrated across 20 wells of a 384-well plate (Corning 3575) using 1.5-fold dilutions with a final volume of 10 μL. Next, 10 μL of PARP1 (5 nM) premixed with fluorescein labeled p18mer DNA (p18mer*, 2.2 nM) was added to the various

dilutions of nucleosomes or DNA, and then incubated for 30 min to ensure complete dissociation of p18mer*. FP using excitation at 482 nm (bandwidth 16 nm), dichroic filter at 496 nm, and emission at 530 nm (bandwidth 40 nm) was measured from the top of the plate using a BMG Labtech CLARIOstar plate reader. The concentration of competitor (nucleosome or free DNA) that yielded a 50% release of p18mer* ($CC_{50}$) was calculated by fitting of the data with a four-parameter binding curve in Prism. All concentrations noted above reflect the final concentration in the plate.

## PARP1 activity as detected by incorporation of $^{32}$P-ADPR using filter plates

PARP1 (30 nM) was pre-incubated with varying concentrations of DNA (0.01–200 nM final) or nucleosomes (0.2–500 nM) and/or HPF1 (2 µM) in assay buffer (50 mM Tris-HCl, pH 8.0, 50 mM NaCl, 1 mM $MgCl_2$, 0.1 mM EDTA, 0.5 mg/mL bovine serum albumin [Ambion]) in 96-well plates (Corning3898). Following addition of $^{32}$P-NAD$^+$ (40 µM, $1.8 \times 10^6$ cpm/well, PerkinElmer) to yield a final volume of 25 µL, reactions were quenched at varying time points (0.33–5 min) by addition of 50 µL of 30% trichloracetic acid (TCA). Samples (50 µL of total) were then loaded onto a Whatman Mini-Fold Spot-Blot apparatus containing a Whatman GF/C glass microfiber filter. Each well was washed three times with 10% TCA (100 µL). After removal of the filter from the apparatus, the filter was gently incubated in 10% TCA (20–40 mL) for three more washes. After drying, the filter was exposed to a Phosphor screen overnight (GE Healthcare) and imaged using a Typhoon FLA 9500 (GE Healthcare). Spot intensities were quantitated using ImageQuant. For converting arbitrary pixel intensities into molar quantities, a known amount of NAD$^+$(4–6 pmol) was blotted onto a filter paper and exposed for the same amount of time. The values for $k_{cat}$ are apparent as they were determined at 40 µM NAD$^+$, which is at the concentration of the $K_m$ for NAD$^+$. The apparent $k_{cat}$ values were determined by fitting the observed incorporation of radioactivity ($cpm_{obs}$) to

$$cpm_{obs} = cpm_{min} + \frac{(cpm_{max} - cpm_{min})}{\left(1 + \left(\frac{[DNA]}{K_{act}}\right)\right)}$$

where $cpm_{min}$ is derived from control samples containing no PARP1, $cpm_{max}$ is the highest observed incorporation, [DNA] is the concentration of DNA or nucleosome, and $K_{act}$ is the concentration of half-maximal activation. The values for $k_{cat}$ determined from $cpm_{max}$ after correction of amount of PARP1 (30 nM) are apparent as they were determined at 40 µM NAD$^+$, which is at the concentration of $K_m$ for NAD$^+$. The values for the apparent activation constant $K_{act}$ were not considered meaningful (i.e., titration of enzyme, not substrate) as they were all at or below the concentration of PARP1 in the assay (30 nM). This was not unexpected, given the previously reported low nM affinity of both free DNA and nucleosome for PARP1 (*Clark et al., 2012*; *Langelier et al., 2008*; *Rudolph et al., 2018*; *Rudolph et al., 2020*).

## PARP1 activity as detected by incorporation of $^{32}$P-ADPR using SDS-PAGE

PARP1 (100 nM) was combined with nucleosome (300–800 nM) and HPF1 (2 µM) in a total volume of 20 µL in assay buffer. PARP1 is activated by nucleosomes, so no further activator was needed. For assays with the H3 peptide substrate (22.5 µM), 100 nM p18mer DNA was included to trigger activation of PARP1. PARylation reactions were initiated by addition of 20 µL of $^{32}$P-NAD$^+$ to a final concentration of 10 µM (~$5 \times 10^6$ cpm/reaction). Aliquots were removed at 10–40 s and quenched by addition of olaparib to 4 µM. For analysis, samples were mixed with Laemmli buffer (2×), heated to 95°C for 3 min, and proteins were separated by SDS-PAGE (4–12% Bis-Tris, Thermo Fisher using 1× MES buffer). The gel was exposed to a PhosphorScreen, and then imaged and quantitated as for the filter-based assay. Signals for PARP1, HPF1, and cumulative histones were quantitated individually and corrected for background using boxes of equivalent area. For converting arbitrary pixel intensities into molar quantities, a known amount of NAD$^+$ (4–6 pmol) was spiked into multiple lanes 10 min after the start of gel electrophoresis for quantitation by ImageQuant.

## FRET assays to measure affinity of HPF1 for PARP1

HPF1 and PARP1 were labeled with Alexa647-$C_2$-maleimide and Alexa488-$C_4$-maleimide, respectively, and their interaction was detected using FRET as previously described (*Rudolph et al., 2021*).

# PARP1 activity as detected by HPLC assays that monitor consumption of NAD$^+$, formation of nicotinamide and ADPR, and release of AMP-PR following digestion with phosphodiesterase

PARP1 (100 nM) was pre-incubated with p18mer or Nuc165 (200 nM) with or without HPF1 (wild-type or E284 mutant, 2 µM) in assay buffer. Reactions were initiated with NAD$^+$ (200 µM) to yield a final volume of 30 µL and were quenched after 1 min with 30 µL of perchloric acid (1 M). After a 15 min incubation on ice, precipitated protein was removed by centrifugation at 14,000×$g$ for 15 min at 4°C. The supernatant was removed and placed into a fresh tube containing 6 µL of sodium acetate (1 M, pH 4.5). This mixture was re-neutralized by addition of 11 µL of KOH (5 N), and the precipitated salts were removed by brief centrifugation at 1000×$g$. The supernatant was removed, loaded into HPLC vials, and analyzed on a Synergi Fusion-RP column (Phenomenex, 150 × 4.6 mm) using the following conditions: start = 97% buffer (20 mM ammonium acetate, pH 4.5), 3% acetonitrile; 10 min gradient to 40% acetonitrile, followed by a 3 min gradient back to 3% acetonitrile; and then a 5 min re-equilibration under those conditions prior to the next injection. ADPR, NAD$^+$, and nicotinamide eluted at 2.8–2.9, 3.9–4, and 5.0–5.1, respectively. The area under the curve was used to determine absolute concentrations by comparison with injection of known amounts of ADPR, NAD$^+$, and nicotinamide using extinction coefficients at 260 nm of 17.4, 13.5, and 2.5 mM$^{-1}$ cm$^{-1}$, respectively.

The HPLC-based assay above was modified to facilitate the direct measurement of chain length extension as follows. Concentrations of p18mer or Nuc165 were increased to 1 µM and were pre-incubated with NAD$^+$ in the presence or absence of HPF1 (wild-type or E284A mutant, 2 µM). Reaction volumes were increased to 450 µL. Following initiation with PARP1 (1 µM), 45 µL aliquots were withdrawn and quenched into 45 µL of perchloric acid (1 M). Precipitation, removal of protein, re-neutralization, and analysis of ADPR, NAD$^+$, and nicotinamide were performed as above. The precipitated protein pellet (containing PARP1, HPF1, and histones) was resuspended in 30 µL of Tris-HCl (100 mM, un-pH'd at ~10) containing 20 mM MgCl$_2$ and 0.6 U/sample of snake venom phosphodiesterase. After incubation for 18 hr at RT, the proteins were precipitated, and AMP-PR$_2$ (branches), AMP-PR (middles), and AMP (ends) were recovered and analyzed by HPLC as above for ADPR, NAD$^+$, and nicotinamide with elution times of 2.0–2.1, 2.9–3.0, and 3.8–3.9 min, respectively. Amounts of AMP-PR$_2$ were too low to quantitate accurately, and amounts of AMP were variable due to NAD$^+$ contamination of protein precipitates prior to digestion. (The remaining NAD$^+$ from the reaction is digested to AMP by phosphodiesterase, and attempts to remove this contaminant by acid wash of the protein pellets prior to digestion led to variable results. We thus limited our analysis to AMP-PR, which reflects the PAR chain length.)

## Acknowledgements

No competing financial interests have been declared. Funding was provided by the National Cancer Institute R01 CA218255 (to KL) and the Howard Hughes Medical Institute (to KL).

## Additional information

### Funding

| Funder | Grant reference number | Author |
| --- | --- | --- |
| National Cancer Institute | CA218255 | Karolin Luger |
| Howard Hughes Medical Institute | | Karolin Luger |
| National Institutes of Health | T32GM008759 | Genevieve Roberts |

The funders had no role in study design, data collection and interpretation, or the decision to submit the work for publication.

## Author contributions
Johannes Rudolph, Conceptualization, Data curation, Formal analysis, Supervision, Funding acquisition, Validation, Investigation, Visualization, Methodology, Writing - original draft, Project administration, Writing - review and editing; Genevieve Roberts, Formal analysis, Investigation, Methodology; Uma M Muthurajan, Investigation, Methodology, Writing - review and editing; Karolin Luger, Conceptualization, Resources, Supervision, Project administration, Writing - review and editing

## Author ORCIDs
Johannes Rudolph https://orcid.org/0000-0003-0230-3323
Uma M Muthurajan https://orcid.org/0000-0003-2596-5848
Karolin Luger https://orcid.org/0000-0001-5136-5331

## Decision letter and Author response
Decision letter https://doi.org/10.7554/eLife.65773.sa1
Author response https://doi.org/10.7554/eLife.65773.sa2

## Additional files

### Supplementary files
• Transparent reporting form

### Data availability
All data generated or analysed during this study are included in the manuscript and supporting files. Source data files have been provided for all Figures.

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
