## [Decision Letter]

**Acceptance summary:**

This manuscript describes a set of biochemical studies on the substrate and reaction specificity of poly(ADP-ribose) polymerase 1 (PARP1), an important antineoplastic drug target and component of DNA damage response. The most significant finding is that histone PARylation factor (HPF1) binding to PARP1 causes a shift from primarily PARylation activity to that of hydrolytic activity, which offers new avenues for understanding and controlling PARP1. The findings described in this paper are of broad interest to readers in the fields of DNA damage response, chromatin structure regulation, and to researchers studying PARP1 and issues related to NAD+ metabolism.

**Decision letter after peer review:**

Thank you for submitting your article "HPF1 and nucleosomes mediate a dramatic switch in activity of PARP1 from polymerase to hydrolase" for consideration by *eLife*. Your article has been reviewed by three peer reviewers, and the evaluation has been overseen by Maria Spies as a Reviewing Editor and Kevin Struhl as the Senior Editor. The reviewers have opted to remain anonymous.

Essential Revisions:

While all reviewers have acknowledged the quality of the experimental work reported in this manuscript, they agreed that the current title and the main message of the manuscript are somewhat provocative and are not fully aligned with the experimental evidence. The authors should thoroughly address the points raised by the reviewers by analyzing the data more critically and tempering the message/title, and/or with new experiments. Specifically:

1) Provide a better explanation of the treadmilling effect. This can be done experimentally, by carrying out the analysis of a mutant at the orthodox carboxylate nucleophile (see Rev. 1 suggestion), providing more serine substrates (see reviewer #2 suggestion), and/or discussing possible alternative mechanisms suggested by reviewers 1 and 2.

2) The results of the gel-based and plate-based assays need to be reconciled. Specifically, no difference in the steady-state kinetic parameters observed in the presence and absence of HPF1 in plate-based assays needs to be reconciled with the different kinetics (burst followed by slow phase) observed in the gel-based assays.

3) Temper the statement regarding the effects of nucleosome vs. DNA interaction on PARP1 (see specific comments by reviewers 1 and 3).

4) Address the comments by reviewers 1 and 3 regarding identification of E284 as a catalytic base.

Reviewer #2 (Recommendations for the authors):

• Using their plate-based assay (which measures total protein ADPr incorporation by PARP1), the authors see no change in Km (for NAD+) or kcat in the presence of HPF1 compared to its absence (Figure 2C) (of note, the HPF1 concentration is missing in the Materials and methods section for this assay). No change in the apparent kcat might be seen as surprising considering the dramatic change in the mechanism, substrate preference, and – as the authors ultimately show – the extent of "non-productive" NAD+ hydrolysis upon HPF1 addition. The raw data in the presence of HPF1 are not shown, but presumably the incorporation is similarly linear to the situation in the absence of HPF1 (Figure 2B), as otherwise it would be commented upon. However, in the gel-based assay, the authors see that HPF1 significantly changes the kinetics of ADPr incorporation for all protein substrates that are present (PARP1, HPF1, and especially histones), resulting in an initial quick burst of modification prior to the first timepoint of 10 sec, followed by slow linear behaviour (Figure 3B). In our view, the burst might correspond to the fast serine ADPribosylation and the linear phase could represent chain extension (very slow because inhibited by HPF1). How are the observations from these two different assays reconciled with each other? Does the rate of ADPr incorporation in the plate-based assay measured in the presence of HPF1 correspond to the burst phase or the linear phase of the gel-based results? If there is a real discrepancy, is it due to technical differences? For example, in case the DNA used contains terminal phosphates, is it conceivable that the filter-based assay measures DNA ADP-ribosylation in addition to protein ADP-ribosylation? Answering this point does not necessarily require new experiments, but more extensive comments on technical aspects of the two assays and how they could be reconciled would be useful.

• The authors write: "That is, although the initiation reaction on serines occurs efficiently, PARP1 then spends most of its catalytic power performing hydrolysis of NAD+, thereby precluding further attachment of ADPR onto protein in the presence of HPF1". This wording suggests a causal link whereby the increased treadmilling revealed with the HPLC-based assay limits the length of synthesised protein-linked PAR chains in the presence of HPF1. However, a reverse causal relationship is also possible and perhaps more likely: because HPF1 binding to PARP1 inhibits chain extension, PARP1 – following mono modification of suitable serine residues – quickly runs out of preferred substrates, and this might increase the chance of "futile" NAD+ hydrolysis due to accepting water as a substrate (in contrast, HPF1-free PARP1 can always extend a growing PAR chain and thus never runs out of a non-water substrate). If this interpretation is correct, would providing more serine substrates, e.g. in the form of a large excess of a suitable serine-containing peptide, limit the observed "futile" NAD+ hydrolysis by PARP1:HPF1 or is treadmilling an inherent property of this complex independent of serine substrate availability? Additionally, it would be useful to comment on the relationship between the HPLC-based assay and the gel-based assay: is the treadmilling measured in what in the gel-based assay corresponds to the burst phase or the linear phase? Perhaps, following the above proposed reasoning, during the burst phase – which might correspond to serine modification – treadmilling is lower and only becomes significant later? In Figure 6C, the first timepoint is after 10 sec, which (if the conditions of these two assays are comparable) would correspond to the burst phase being already over. The authors make a reference to the gel-based assay in one sentence ("The observed dramatic increase in treadmilling explains the lack of linearity (i.e. the burst in PARylation) seen in the gel-based assays with HPF1 above") but they do so in a way that still leaves it fairly unclear what is the cause and what is the effect. If possible, it might be useful to address objections raised here experimentally (e.g. by performing the experiment in the presence of an increased amount of a serine substrate) and/or with a more extensive discussion.

Reviewer #3 (Recommendations for the authors):

The manuscript "HPF1 and nucleosomes mediate a dramatic switch in activity of PARP1 from polymerase to Hydrolase" by Rudolph et al. studies the effect of HPF1 on the steps of the catalytic reaction of PARP1. They use various PARP1 activators i.e. free DNA and varied forms of core nucleosomes to quantify reaction rates in the presence and absence of HPF1, using several assays. The main point of the manuscript is the observation that in the presence of HPF1, PARP1 is converted to an NAD+ hydrolase, which releases free ADPr, instead of its normal activity to produce ADPr polymers. The PARP1 hydrolase activity has been described previously, but they now show that HPF1 increases it substantially under the conditions that they tested. The authors also describe their independent identification of HPF1 residue E284 as a residue that is essential for Ser modification, confirming previous structural and biochemical work from Ivan Ahel's group. Although the assays are well performed and controlled and yield important quantitative information that was missing in the field, the main result of the hydrolase activity of PARP1 is hard to reconcile with current knowledge of HPF1 effects in cell-based experiments.

Here are some specific points that would need to be addressed before publication of the results:

The authors report that in the presence of HPF1, PARP1 spends most of its catalytic power (about 90%) consuming NAD+ to form free ADPr. However, it has been shown that knocking out HPF1 in cells leads to a dramatic reduction in PAR production in the presence of DNA damage (Palazzo et al., 2018), indicating that HPF1 is necessary for a large portion of the PAR produce by PARP1 during DNA damage. The vast majority of the PAR produced in response to DNA damage is Ser linked (ibid), so requiring HPF1 activity. If HPF1's role is to convert PARP1 into a hydrolase, rather than a PAR polymerase, how can these results be explained? Can the authors address this apparent discrepancy in HPF1 function?

"We conclude that there is a unique aspect of the PARP1 – nucleosome interaction compared to the PARP1 – free DNA interaction that leads to more robust autoPARylation,…" According to Table 1, the increase in kcat is less than two-fold. I would recommend rephrasing and mentioning a modest increase.

Discussion, "We began our work by demonstrating that nucleosomes are better activators of PARP1 than free DNA (Figure 2, Table 1)." Same as above.

"Perhaps there are direct interactions between the unfolded (or alternatively folded) HD-domain and nucleosomes that stabilize the open conformation and prevent this domain from collapsing back onto the catalytic domain." Is there anything in the cryo-EM structure of PARP2/HPF1/nucleosome that would support the hypothesis that nucleosomes directly contact the HD?

"located primarily in the BRCT". The most recent mass spec studies identify the modified residues to be located outside of the BRCT domain, in the linker region between the BRCT and the WGR domain. Perhaps it is better to refer to the automodification region, and not the BRCT.

Although it is nice that they figured out independently that E284 of HPF1 residue is necessary for the catalysis of Ser modification, I am wondering if it justified to spend this amount of time/space describing the mutagenesis work? The reported work does not in itself identify E284 as the catalytic base. The structure of the HPF1-PARP2 complex and the positioning of E284 in the PARP active site is really the convincing data. The claim that the presented work identifies E284 as the catalytic base is not quite right.

"Consistent with Glu284 on HPF1, not Glu988 on PARP1, performing the de-protonation required for initiation, we (data not shown) and Ahel's group observe HPF1-mediated incorporation of ADPR by the E988K mutant of PARP1, albeit at significantly slower rates." This data should be shown, or just reference the Ahel study.

It will be important to analyze the HPF1 effect on PARP2.

---

## [Author Response]

Essential Revisions:While all reviewers have acknowledged the quality of the experimental work reported in this manuscript, they agreed that the current title and the main message of the manuscript are somewhat provocative and are not fully aligned with the experimental evidence. The authors should thoroughly address the points raised by the reviewers by analyzing the data more critically and tempering the message/title, and/or with new experiments. Specifically:1) Provide a better explanation of the treadmilling effect. This can be done experimentally, by carrying out the analysis of a mutant at the orthodox carboxylate nucleophile (see Rev. 1 suggestion), providing more serine substrates (see reviewer #2 suggestion), and/or discussing possible alternative mechanisms suggested by reviewers 1 and 2.

We do not believe it is possible to remove all the orthodox carboxylate nucleophiles (>100) as described below. We have performed the serine-titration experiment suggested by Rev. 2 and incorporated these results into the manuscript.

2) The results of the gel-based and plate-based assays need to be reconciled. Specifically, no difference in the steady-state kinetic parameters observed in the presence and absence of HPF1 in plate-based assays needs to be reconciled with the different kinetics (burst followed by slow phase) observed in the gel-based assays.

We have addressed these discrepancies. See below.

3) Temper the statement regarding the effects of nucleosome vs. DNA interaction on PARP1 (see specific comments by reviewers 1 and 3).

See below. We point to the statistical evaluation of this difference and stand by our conclusions.

4) Address the comments by reviewers 1 and 3 regarding identification of E284 as a catalytic base.

Done. See below.

Reviewer #2 (Recommendations for the authors):• Using their plate-based assay (which measures total protein ADPr incorporation by PARP1), the authors see no change in Km (for NAD+) or kcat in the presence of HPF1 compared to its absence (Figure 2C) (of note, the HPF1 concentration is missing in the Materials and methods section for this assay).

2 µM. Now added to the Materials and methods.

No change in the apparent kcat might be seen as surprising considering the dramatic change in the mechanism, substrate preference, and – as the authors ultimately show – the extent of "non-productive" NAD+ hydrolysis upon HPF1 addition. The raw data in the presence of HPF1 are not shown, but presumably the incorporation is similarly linear to the situation in the absence of HPF1 (Figure 2B), as otherwise it would be commented upon. However, in the gel-based assay, the authors see that HPF1 significantly changes the kinetics of ADPr incorporation for all protein substrates that are present (PARP1, HPF1, and especially histones), resulting in an initial quick burst of modification prior to the first timepoint of 10 sec, followed by slow linear behaviour (Figure 3B). In our view, the burst might correspond to the fast serine ADPribosylation and the linear phase could represent chain extension (very slow because inhibited by HPF1). How are the observations from these two different assays reconciled with each other? Does the rate of ADPr incorporation in the plate-based assay measured in the presence of HPF1 correspond to the burst phase or the linear phase of the gel-based results? If there is a real discrepancy, is it due to technical differences? For example, in case the DNA used contains terminal phosphates, is it conceivable that the filter-based assay measures DNA ADP-ribosylation in addition to protein ADP-ribosylation? Answering this point does not necessarily require new experiments, but more extensive comments on technical aspects of the two assays and how they could be reconciled would be useful.

We thank the reviewer for their excellent suggestion of doing a more direct comparison between the plate-based and gel-based assays and have two points that help correlate these two methods.

1) We have added Figure 2—figure supplement 1, that shows the significantly reduced linearity of ADPR-incorporation in the presence of HPF1. As we should have noted previously, but do now (Results), the linear phase is shorter in the presence of HPF1 (0 – 100 s to achieve an R2 > 0.96) vs. in the absence of HPF1 (0 – 150 s to achieve an R2 > 0.97). Additionally, the fitted lines in the assay in the presence of HPF1 do not go through (0,0) as nicely as those in the absence of HPF1, showing a slight burst. This behavior correlates with the gel-based assay described in Figure 3B.

2) The quench method used in the gel-based assay (addition of olaparib to 4 µM), is presumably not as “instant” as the acid quench used in the plate-based assay. Thus, the short time points we use may be susceptible to a lag in quenching that may be detrimental to achieving true linear rates. We have added a comment about this in the Results section.

• The authors write: "That is, although the initiation reaction on serines occurs efficiently, PARP1 then spends most of its catalytic power performing hydrolysis of NAD+, thereby precluding further attachment of ADPR onto protein in the presence of HPF1". This wording suggests a causal link whereby the increased treadmilling revealed with the HPLC-based assay limits the length of synthesised protein-linked PAR chains in the presence of HPF1. However, a reverse causal relationship is also possible and perhaps more likely: because HPF1 binding to PARP1 inhibits chain extension, PARP1 – following mono modification of suitable serine residues – quickly runs out of preferred substrates, and this might increase the chance of "futile" NAD+ hydrolysis due to accepting water as a substrate (in contrast, HPF1-free PARP1 can always extend a growing PAR chain and thus never runs out of a non-water substrate). If this interpretation is correct, would providing more serine substrates, e.g. in the form of a large excess of a suitable serine-containing peptide, limit the observed "futile" NAD+ hydrolysis by PARP1:HPF1 or is treadmilling an inherent property of this complex independent of serine substrate availability?

This is an excellent idea and we thank the reviewer for their helpful insight. We have performed this experiment using p18mer as an activator, titrating in increasing concentrations of the H3 tail peptide as a substrate for serine-PARylation. As shown in the new Figure 5D, providing more serine PARylation sites does in fact inhibit the treadmilling effect down to baseline levels, thus increasing the overall amount of protein-linked ADPR. Therefore, we agree with the reviewer that the causal link should be expressed as follows: HPF1 blocks chain extension, and thus PARP1, still bound to activating DNA (i.e. nucleosome), quickly runs out of suitable sites for PARylation and therefore uses water as a nucleophile leading to the treadmilling effect. We have added verbiage to both the Results and the Discussion.

To be noted, we ruled out performing this experiment by titrating in nucleosomes for two reasons. (1) Changing nucleosome concentrations changes both the DNA and the serine-substrate concentrations, complicating any interpretation. (2) Higher concentrations of nucleosome (> 1 µM) leads to protein precipitation in the presence of PARP1.

Additionally, it would be useful to comment on the relationship between the HPLC-based assay and the gel-based assay: is the treadmilling measured in what in the gel-based assay corresponds to the burst phase or the linear phase? Perhaps, following the above proposed reasoning, during the burst phase – which might correspond to serine modification – treadmilling is lower and only becomes significant later? In Figure 6C, the first timepoint is after 10 sec, which (if the conditions of these two assays are comparable) would correspond to the burst phase being already over. The authors make a reference to the gel-based assay in one sentence ("The observed dramatic increase in treadmilling explains the lack of linearity (i.e. the burst in PARylation) seen in the gel-based assays with HPF1 above") but they do so in a way that still leaves it fairly unclear what is the cause and what is the effect. If possible, it might be useful to address objections raised here experimentally (e.g. by performing the experiment in the presence of an increased amount of a serine substrate) and/or with a more extensive discussion.

We have attempted to compare the ratio of treadmilling as a function of extent of reaction and seen no differences as summarized in the Discussion and seen in Figure 5C vs. Figure 6C. We have attempted to quantitate these effects at really short reaction times (10 sec). However, the very small amounts of both ADPR and nicotinamide formed preclude accurate ratio calculations.

As noted for the previous point, we have added the experiment with the serine substrate titration (new Figure 5D) and added to the Discussion about these results.

Reviewer #3 (Recommendations for the authors):The manuscript "HPF1 and nucleosomes mediate a dramatic switch in activity of PARP1 from polymerase to Hydrolase" by Rudolph et al. studies the effect of HPF1 on the steps of the catalytic reaction of PARP1. They use various PARP1 activators i.e. free DNA and varied forms of core nucleosomes to quantify reaction rates in the presence and absence of HPF1, using several assays. The main point of the manuscript is the observation that in the presence of HPF1, PARP1 is converted to an NAD+ hydrolase, which releases free ADPr, instead of its normal activity to produce ADPr polymers. The PARP1 hydrolase activity has been described previously, but they now show that HPF1 increases it substantially under the conditions that they tested. The authors also describe their independent identification of HPF1 residue E284 as a residue that is essential for Ser modification, confirming previous structural and biochemical work from Ivan Ahel's group. Although the assays are well performed and controlled and yield important quantitative information that was missing in the field, the main result of the hydrolase activity of PARP1 is hard to reconcile with current knowledge of HPF1 effects in cell-based experiments.Here are some specific points that would need to be addressed before publication of the results:The authors report that in the presence of HPF1, PARP1 spends most of its catalytic power (about 90%) consuming NAD+ to form free ADPr. However, it has been shown that knocking out HPF1 in cells leads to a dramatic reduction in PAR production in the presence of DNA damage (Palazzo et al., 2018), indicating that HPF1 is necessary for a large portion of the PAR produce by PARP1 during DNA damage.

The large reduction in activity seen for the HPF1-/- cell line in Palazzo et al. is in pan-ADP-ribosylation (Figure 1A, left panel). Overall PARylation levels are quite similar between WT and the HPF1-/- cell line as detected using anti-PAR (Figure 1A, right panel).

The vast majority of the PAR produced in response to DNA damage is Ser linked (ibid), so requiring HPF1 activity.

The results we present in this manuscript agree with this previous observation. See Figure 3D.

If HPF1's role is to convert PARP1 into a hydrolase, rather than a PAR polymerase, how can these results be explained? Can the authors address this apparent discrepancy in HPF1 function?

We make a few comments here and in the manuscript that we hope are helpful. We do not dispute that significant PARylation does occur in the presence of HPF1. However, we do demonstrate the presence of HPF1 leads to much shorter PAR chains (in agreement with previous results; see Gibbs-Seymour et al., 2016 and Ando et al., 2019) and that these shorter chains arise because of the treadmilling effect (see Figure 5 and Figure 6 and the Discussion).

1) Our results showing similar levels of incorporation of ADPR into proteins in the presence or absence of HPF1 (Figure 2C and Figure 3A) agree qualitatively with in vitro assays monitoring incorporation of ^32^P-ADPR by Gibbs-Seymour et al. We have added a comparison of our data to Gibbs-Seymour et al., 2016 in the Results.

2) Just as Palazzo et al. see similar levels of PAR in WT vs. HPF1-/- cell lines, we see similar levels of total protein PARylation in the presence and absence of HPF1 (see Figure 2C). We have added a comparison of our data to Palazzo et al., 2018 in the Results.

"We conclude that there is a unique aspect of the PARP1 – nucleosome interaction compared to the PARP1 – free DNA interaction that leads to more robust autoPARylation,…" According to Table 1, the increase in kcat is less than two-fold. I would recommend rephrasing and mentioning a modest increase.

See response to reviewer #1 regarding the statistical significance of these data.

Discussion, "We began our work by demonstrating that nucleosomes are better activators of PARP1 than free DNA (Figure 2, Table 1)." Same as above.

See response to reviewer #1 regarding the statistical significance of these data.

"Perhaps there are direct interactions between the unfolded (or alternatively folded) HD-domain and nucleosomes that stabilize the open conformation and prevent this domain from collapsing back onto the catalytic domain." Is there anything in the cryo-EM structure of PARP2/HPF1/nucleosome that would support the hypothesis that nucleosomes directly contact the HD?

The reviewer makes an excellent suggestion. Yes, the loop in the HD helices (residues L302 and R303) contact the linker DNA. We have added a comment about this to the Discussion.

"located primarily in the BRCT". The most recent mass spec studies identify the modified residues to be located outside of the BRCT domain, in the linker region between the BRCT and the WGR domain. Perhaps it is better to refer to the automodification region, and not the BRCT.

Good suggestion; changed.

Although it is nice that they figured out independently that E284 of HPF1 residue is necessary for the catalysis of Ser modification, I am wondering if it justified to spend this amount of time/space describing the mutagenesis work? The reported work does not in itself identify E284 as the catalytic base. The structure of the HPF1-PARP2 complex and the positioning of E284 in the PARP active site is really the convincing data. The claim that the presented work identifies E284 as the catalytic base is not quite right.

We strongly disagree. Showing that something is in proximity alone is not the defining feature of a catalytic base. Showing that its removal leads to loss of activity, but not loss of binding, that is the key experiment that demonstrates its direct role in catalysis.

"Consistent with Glu284 on HPF1, not Glu988 on PARP1, performing the de-protonation required for initiation, we (data not shown) and Ahel's group observe HPF1-mediated incorporation of ADPR by the E988K mutant of PARP1, albeit at significantly slower rates." This data should be shown, or just reference the Ahel study.

We refer to the Ahel study alone as we performed the experiment only once to “spot check” whether we agreed with their results.

It will be important to analyze the HPF1 effect on PARP2.

We have realized this as well and intend to let the next graduate student sort out these potentially interesting new discoveries.